# ACCELERATING FIRST-ORDER METHODS FOR BILEVEL OPTIMIZATION UNDER GENERAL SMOOTHNESS

## ABSTRACT

Bilevel optimization is pivotal in machine learning applications such as hyperparameter tuning and adversarial training. While existing methods for nonconvex-strongly-convex bilevel optimization can find an $\epsilon$-stationary point under Lipschitz continuity assumptions, two critical gaps persist: improving algorithmic complexity and generalizing smoothness conditions. This paper addresses these challenges by introducing an accelerated framework under Hölder continuity—a broader class of smoothness that subsumes Lipschitz continuity. We propose a restarted accelerated gradient method that leverages inexact hypergradient estimators and establishes theoretical oracle complexity for finding $\epsilon$-stationary points. Empirically, experiments on data hypercleaning and hyperparameter optimization demonstrate superior convergence rates compared to state-of-the-art baselines.

## 1 INTRODUCTION

Bilevel optimization is a powerful paradigm with applications in various machine learning tasks, such as hyperparameter tuning [1; 2; 3], adversarial training [4; 5; 6; 7], and reinforcement learning [8; 9; 10]. It involves two levels of optimization, where the objective at the upper level depends on the solution to a lower-level optimization problem. The general bilevel problem can be expressed as:

$$\min_{x \in \mathbb{R}^{d_x}, y \in Y^*(x)} f(x, y), \quad \text{where } Y^*(x) = \arg\min_{y \in \mathbb{R}^{d_y}} g(x, y). \tag{1}$$

In this formulation, $f(x, y)$ denotes the upper-level objective, while $g(x, y)$ denotes the lower-level objective.

This study examines the nonconvex-strongly-convex framework, wherein the lower-level function $g(x, y)$ exhibits strong convexity with respect to $y$, while the upper-level function $f(x)$ is possibly nonconvex. In this case, the lower-level objective admits a unique solution $Y^*(x) = \{y^*(x)\}$. Then Problem equation 1 is equivalent to minimizing the hyper-objective function

$$\varphi(x) := f(x, y^*(x)), \quad \text{where } y^*(x) = \arg\min_{y \in \mathbb{R}^{d_y}} g(x, y).$$

As shown in [11; 12], the hyper-gradient $\nabla\varphi(x)$ is given by:

$$\begin{aligned}
\nabla\varphi(x) &= \nabla_x f(x, y) + \nabla y^*(x)\nabla_y f(x, y^*(x)) \\
&= \nabla_x f(x, y^*(x)) - \nabla^2_{xy} g(x, y^*(x)) \left[\nabla^2_{yy} g(x, y^*(x))\right]^{-1} \nabla_y f(x, y^*(x)).
\end{aligned} \tag{2}$$

The goal of this paper is to find the point $x$ such that $\varphi(x)$ is an $\epsilon$-stationary point, i.e., $\|\nabla\varphi(x)\| \le \epsilon$. For nonconvex-strongly-convex bilevel optimization, previous work [13; 14; 15] primarily focuses on assuming Lipschitz continuity of $\nabla f$, $\nabla g$, $\nabla^2 g$, and $\nabla^3 g$, and either approximates the hyper-gradient $\nabla\varphi(x)$ or minimizes a penalty function. Approximating the hyper-gradient $\nabla\varphi(x)$ requires first-order oracle access to $f$ and second-order oracle access to $g$, whereas minimizing the penalty function only requires first-order oracle access to both $f$ and $g$.

Two key open questions remain: (i) For first-order methods, it remains open whether the existing algorithmic complexities for finding approximate first-order stationary points in nonconvex–strongly-convex bilevel optimization can be further improved under high order smoothness, and (ii) whether the Lipschitz continuity assumptions can be generalized to the Hölder continuity.

## 1.1 Related Work

**Nonconvex optimization:** For unconstrained nonconvex objectives with Lipschtiz continuous gradient, the classical gradient descent (GD) is known to find an $\epsilon$-stationary point within $\mathcal{O}(\epsilon^{-2})$ gradient computations [16]. This rate is optimal among the first-order methods [17; 18]. Under the additional assumption of Lipschitz continuous Hessians, accelerated gradient descent (AGD) [19; 20; 21] finds an $\epsilon$-stationary point in $\tilde{\mathcal{O}}(\epsilon^{-7/4})$ evaluations. [22] and [23] further show that AGD with restarts achieves $\mathcal{O}(\epsilon^{-7/4})$ complexity for finding $\epsilon$-stationary points, without additional log factors. Under the more general assumption of Hölder continuity of the Hessian, [24] proposed a universal, parameter-free heavy-ball method equipped with two restart mechanisms, achieving a complexity bound of $\mathcal{O}(H_\nu^{1/(2+2\nu)} \epsilon^{-(4+3\nu)/(2+2\nu)})$ in terms of function and gradient evaluations, where $\nu \in [0, 1]$ and $H_\nu$ denote the Hölder exponent and constant, respectively.

**Bilevel Optimization Methods:** To approximate the hyper-gradient, gradient-based methods contain approximate implicit differentiation (AID) [25; 11; 26; 27; 11] and iterative differentiation (ITD) [25; 11; 26; 11; 28]. Using the hyper-gradient equation 2, one can find an $\epsilon$-stationary point of $\varphi(x)$ within $\tilde{\mathcal{O}}(\epsilon^{-2})$ first-order oracle calls from $f$ and $\tilde{\mathcal{O}}(\epsilon^{-2})$ second-order oracle calls from $g$ [29; 26]. In practical implementations, these methods typically rely on access to Jacobian or Hessian-vector product oracles. [14] proposed a fully first-order method that does not require Jacobian or Hessian-vector product oracles, and finds an $\epsilon$-stationary point using only first-order gradients of $f$ and $g$. Concurrently, [13] proposed a method that achieves a near-optimal convergence rate of $\tilde{\mathcal{O}}(\epsilon^{-2})$. Moreover, under high-order smoothness assumptions, they established an accelerated convergence rate of $\tilde{\mathcal{O}}(\epsilon^{-7/4})$.

Table 1: Complexity bounds for finding $\epsilon$-stationary points under Lipschitz continuity assumptions.

| Algorithm | Gc($f, \epsilon$) | Gc($g, \epsilon$) | JV($g, \epsilon$) | HV($g, \epsilon$) |
|---|---|---|---|---|
| AID-BiO ([26]) | $\mathcal{O}(\kappa^3 \epsilon^{-2})$ | $\mathcal{O}(\kappa^3 \epsilon^{-2})$ | $\mathcal{O}(\kappa^3 \epsilon^{-2})$ | $\tilde{\mathcal{O}}(\kappa^3 \epsilon^{-2})$ |
| ITD-BiO ([26]) | $\mathcal{O}(\kappa^3 \epsilon^{-2})$ | $\mathcal{O}(\kappa^4 \epsilon^{-2})$ | $\tilde{\mathcal{O}}(\kappa^4 \epsilon^{-2})$ | $\tilde{\mathcal{O}}(\kappa^4 \epsilon^{-2})$ |
| RAHGD ([15]) | $\tilde{\mathcal{O}}(\ell^{3/4} \kappa^{11/4} \epsilon^{-7/4})$ | $\tilde{\mathcal{O}}(\ell^{3/4} \kappa^{13/4} \epsilon^{-7/4})$ | $\tilde{\mathcal{O}}(\ell^{3/4} \kappa^{11/4} \epsilon^{-7/4})$ | $\tilde{\mathcal{O}}(\ell^{3/4} \kappa^{13/4} \epsilon^{-7/4})$ |
| F²BA([13]) | $\tilde{\mathcal{O}}(\ell \kappa^4 \epsilon^{-2})$ | $\tilde{\mathcal{O}}(\ell \kappa^4 \epsilon^{-2})$ | \ | \ |
| AccF²BA([13]) | $\tilde{\mathcal{O}}(\ell^{3/4} \kappa^{13/4} \epsilon^{-7/4})$ | $\tilde{\mathcal{O}}(\ell^{3/4} \kappa^{13/4} \epsilon^{-7/4})$ | \ | \ |
| Proposed method (this work) | $\tilde{\mathcal{O}}(\ell^{3/4} \kappa^{13/4} \epsilon^{-7/4})$ | $\tilde{\mathcal{O}}(\ell^{3/4} \kappa^{13/4} \epsilon^{-7/4})$ | \ | \ |

## 1.2 Our Contribution

In this paper, we propose an accelerated first-order algorithm for solving nonconvex–strongly convex bilevel optimization problems. Our main contributions are summarized as follows:

1. We introduce an accelerated first-order method framework—originally developed for nonconvex optimization—into the setting of nonconvex–strongly convex bilevel optimization, and consider more general Hölder continuity assumptions on $f$ and $g$.

2. We prove that, with a carefully designed restart condition, the iterates generated by our proposed method remain uniformly bounded within each epoch. Based on this, we demonstrate that the algorithm is convergent with accelerated performance.

3. Even under the standard Lipschitz continuity setting, our method improves the first-order oracle complexity for finding an $\epsilon$-stationary point of $\varphi(x)$ to $\tilde{\mathcal{O}}(\ell^{3/4} \kappa^{13/4} \epsilon^{-7/4})$, without requiring access to second-order oracles, where $\ell$ and $\kappa$ denote the problem's largest smoothness and condition number. This bound improves upon previously known results, as summarized in Table 1, and is consistent with the concurrent findings of [13], who established a similar $\tilde{\mathcal{O}}(\epsilon^{-7/4})$ rate under a different restarting scheme.

4. Our experimental results further support the theoretical convergence guarantees.

**Organization.** The rest of this work is organized as follows. Section 2 delineates the assumptions and specific algorithmic subroutines. Section 3 formally presents our proposed algorithm along with some basic lemmas. Section 4 provides a complexity bound for finding approximate first-order stationary points. In Section 5, we provide some numerical experiments to show the outstanding performance of our proposed method. Section 6 concludes the paper and discusses future directions. Technical analyses are deferred to the appendix.

**Notation.** Let $a, b \in \mathbb{R}^d$ be vectors, where $\langle a, b \rangle$ represents their inner product and $\|a\|$ denotes the Euclidean norm. For a matrix $A \in \mathbb{R}^{m \times n}$, $\|A\|$ is used to denote the operator norm, which is equivalent to the largest singular value of the matrix. Let $Gc(f, \epsilon)$ and $Gc(g, \epsilon)$ denote the number of gradient evaluations with respect to $f$ and $g$, respectively. Let $JV(g, \epsilon)$ denote the number of Jacobian-vector products $\nabla^2_{xy} g(x, y) v$, and $HV(g, \epsilon)$ denote the number of Hessian-vector products $\nabla^2_{yy} g(x, y) v$. The diameter $\mathcal{R}$ of a compact set $C$ is defined as $\mathcal{R} := \max_{x_1, x_2 \in C} \|x_1 - x_2\|$.

## 2 PRELIMINARIES

In this section, we present the key definitions and assumptions used throughout the paper.

**Definition 1** (Restricted Hölder Continuity). *Let $h$ be a twice differentiable function. We say that $\nabla^2 h$ is restrictively $(\nu, H_\nu)$-Hölder continuous with diameter $\mathcal{R} > 0$ if*

$$H_\nu := \sup_{\|x-y\| \leq \mathcal{R}} \frac{\|\nabla^2 h(x) - \nabla^2 h(y)\|}{\|x - y\|^\nu} < +\infty, \quad \nu \in [0, 1].$$

*When $\mathcal{R} = +\infty$, we call $\nabla^2 h$ is $(\nu, H_\nu)$-Hölder continuous if $\nu \in [0, 1]$ and $H_\nu < +\infty$.*

We make the following assumptions on the upper-level function $f$ and lower-level function $g$:

**Assumption 1.** *We make the following assumptions:*

  i. *The function $\varphi(x)$ is lower bounded.*

  ii. *The function $g(x, y)$ is $\mu$-strongly convex in $y$, and has $L_g$-Lipschitz continuous gradients.*

  iii. *The function $g(x, y)$ has $\rho_g$-Lipschitz continuous Hessians and is $(\nu_g, M_g)$-Hölder continuous in its third-order derivatives.*

  iv. *The function $f(x, y)$ is $C_f$-Lipschitz continuous in $y$ and has $L_f$-Lipschitz continuous gradients.*

  v. *The Hessian $\nabla^2_{xx} f(x, y)$ is $(\nu_f, H_f)$-Hölder continuous.*

  vi. *The mixed and second-order partial derivatives $\nabla^2_{xy} f(x, y)$, $\nabla^2_{yx} f(x, y)$, and $\nabla^2_{yy} f(x, y)$ are $\rho_f$-Lipschitz continuous.*

The assumptions employed in this study are consistent with those commonly adopted in prior literature [13; 27; 14; 15]. To introduce Hölder continuity, we extend the Lipschitz continuity assumptions about the Hessian of $f$, and the third-order derivative of $g$ to our assumptions equation iii, equation v, equation vi.

**Definition 2.** *Under Assumption 1, we define the largest smoothness constant as*

$$\ell := \max \left\{ C_f, L_f, H_f, \rho_f, L_g, \rho_g, M_g \right\},$$

*and the condition number as $\kappa := \ell / \mu$.*

Observe that problem equation 1 can be reformulated as:

$$\min_{x \in \mathbb{R}^{d_x}, \, y \in \mathbb{R}^{d_y}} f(x, y^*(x)), \quad \text{s.t. } g(x, y) - g^*(x) \leq 0, \tag{3}$$

where $g^*(x) = g(x, y^*(x))$ is the value function. A nature penalty problem associated with problem equation 3 is

$$\min_{x \in \mathbb{R}^{d_x}, \, y \in \mathbb{R}^{d_y}} L_\lambda(x, y) := f(x, y) + \lambda \left( g(x, y) - g^*(x) \right),$$

where $\lambda > 0$ is a penalty parameter. This problem is equivalent to minimizing the following auxiliary function:

$$L_\lambda^*(x) := L_\lambda\left(x, y_\lambda^*(x)\right), \text{ where } y_\lambda^*(x) = \arg\min_{y \in \mathbb{R}^d} L_\lambda(x, y). \tag{4}$$

It has been proven in [13] that $L_\lambda^*(x)$ and $\nabla L_\lambda^*(x)$ asymptotically approximate $\varphi(x)$ and $\nabla\varphi(x)$, respectively, as $\lambda$ is sufficiently large. Moreover, $\nabla L_\lambda^*(x)$ is Lipschitz continuous and its Lipschitz constant does not involve $\lambda$. We restate their result below for completeness.

**Lemma 1** ([13, Lemma 4.1]). *Under Assumption 1, for $\lambda \geq 2L_f/\mu$, we have*

  *i.* $|L_\lambda^*(x) - \varphi(x)| \leq \mathcal{O}(\ell\kappa^2/\lambda)$,

  *ii.* $\|\nabla L_\lambda^\star(x) - \nabla\varphi(x)\| \leq \mathcal{O}(\ell\kappa^3/\lambda)$,

  *iii.* $\nabla L_\lambda^\star(x)$ *is* $\mathcal{O}(\ell\kappa^3)$-*Lipschitz continuous.*

In the remainder of the article, we denote the Lipshitz continuous constant of $\nabla L_\lambda^*(x)$ in Lemma 1 by $L = \mathcal{O}(\ell\kappa^3)$ for convenience. Then we introduce a lemma showing that $\nabla^2 L_\lambda^*(x)$ is restrictively $(\nu_f, H_\nu)$-Hölder continuous with diameter $\mathcal{R}$, where the detailed expression of $H_\nu$, depending on $\lambda$ and $\mathcal{D}$, can be found in equation 16 of Appendix C.1.

**Lemma 2.** *Under Assumption 1, for $\lambda \geq 2L_f/\mu$, $\nabla^2 L_\lambda^\star(x)$ is restrictly $(\nu_f, H_\nu(\lambda, \mathcal{R}))$-Hölder continuous with diameter $\mathcal{R} > 0$, where*

$$H_\nu(\lambda, \mathcal{R}) = \mathcal{O}(\ell\kappa^{\nu_f}) + \mathcal{O}(\lambda^{1-\nu_g}\ell\kappa^{4+\nu_g})\mathcal{R}^{1-\nu_f}.$$

## 3 RESTARTED ACCELERATED GRADIENT DESCENT UNDER GENERAL SMOOTHNESS

In this section, we present our algorithm in Algorithm 1 and discuss several of its key properties. The algorithm has a nested loop structure. The outer loop uses the accelerated gradient descent (AGD) method with a restart schemes, inspired from the recently works in [22; 23]. The iteration counter $k$ is reset to 0 when AGD restarts, whereas the total iteration counter $K$ is not. We refer to the period between a reset of $k$ and the next reset as an epoch. We introduce a subscript $t$ to denote the number of restarts. It is important to note that the subscript $t$ in Algorithm 1 is primarily included to facilitate a simpler convergence analysis. Provided that no ambiguity occurs, we omit the subscript $t$, which means that the iterates are within the same epoch.

In Lines 4 and 5, we invoke AGD, which is summarized in Algorithm 2, to find estimators of $y^*(w_{t,k})$ and $y_\lambda^*(w_{t,k})$, respectively. AGD achieves linear convergence when applied to the minimization of smooth and strongly convex functions $g(x, \cdot)$ and $f(x, \cdot) + \lambda g(x, \cdot)$. We note that the iteration number of inner AGD steps plays an important role in the complexity analysis. We will provide the parameters setting for AGD subroutines in Section 4. In the following, we describe some operations involved in the algorithm.

**Restart Condition.** Here, we focus on the iterates within a single epoch and omit the subscript $t$, which indexes different epochs. Then we define $S_k = \sum_{i=1}^k \|x_i - x_{i-1}\|^2$, and the restart condition

$$(k+1)^{4+\nu_f} H_\nu^2 S_k^{\nu_f} > L^2, \tag{5}$$

where the constant $H_\nu$ will be defined in equation 6 below. If equation 5 holds, the epoch terminates; otherwise, it continues. We say that an epoch ends at iteration $k$, if $S_k$ triggers the restart condition equation 5. It is worth noting that, unlike the restart conditions in [22; 15] and the concurrent work by [13], our restart condition is independent of $\epsilon$.

**Hölder Constant $H_\nu$.** From Lemma 2, $\nabla^2 L_\lambda^\star(x)$ is restrictively $(\nu_f, H_\nu(\lambda, \mathcal{R}))$-Hölder continuous with diameter $\mathcal{R} > 0$. Here we choose a specific $\mathcal{R}$ and the corresponding $H_\nu(\lambda, \mathcal{R})$, denoted by $\mathcal{D}$ and $H_\nu$, satisfying

$$\mathcal{D} = \mathcal{O}\left(\lambda^{-(1-\nu_g)}\kappa^{-(1+\nu_g)}\right), \quad H_\nu = \mathcal{O}\left(\lambda^{\nu_f(1-\nu_g)}\ell\kappa^{3+(1+\nu_g)\nu_f}\right). \tag{6}$$

The derivation of $H_\nu$ and $\mathcal{D}$ is provided in equation 18 of Appendix D. Then $\nabla^2 L_\lambda^*(x)$ is restrictively $(\nu_f, H_\nu)$-Hölder continuous with diameter $\mathcal{D}$. In the case of Lipschitz continuity, i.e., $\nu_f = \nu_g = 1$, equation 6 implies $H_\nu = \mathcal{O}(\ell\kappa^5)$ and $\mathcal{D} = \mathcal{O}(\kappa^{-2})$.

---

**Algorithm 1** Restarted Accelerated gradient descent under General Smoothness (RAGD-GS)

---

1: **Input:** initial point $x_{0,0}$; gradient Lipschitz constant $L > 0$; Hessian Hölder constant $H_\nu > 0$ and $\nu_f \in [0, 1]$; penalty parameter $\lambda > 0$; momentum parameter $\theta_k \in (0, 1)$; parameters $\alpha, \alpha' > 0, \beta, \beta' \in (0, 1), \{T_{t,k}\}, \left\{T'_{t,k}\right\}$ of AGD

2: $k \leftarrow 0, K \leftarrow 0, t \leftarrow 0, w_{0,0} \leftarrow x_{0,0}, y_{0,-1} \leftarrow 0, z_{0,-1} \leftarrow 0$

3: **repeat**

4:     $z_{t,k} \leftarrow \text{AGD}\left(g\left(w_{t,k}, \cdot\right), z_{t,k-1}, T_{t,k}, \alpha, \beta\right)$

5:     $y_{t,k} \leftarrow \text{AGD}\left(f\left(w_{t,k}, \cdot\right) + \lambda g\left(w_{t,k}, \cdot\right), y_{t,k-1}, T'_{t,k}, \alpha', \beta'\right)$

6:     $\hat{\nabla} L_\lambda^*(w_{t,k}) \leftarrow \nabla_x f\left(w_{t,k}, y_{t,k}\right) + \lambda\left(\nabla_x g\left(w_{t,k}, y_{t,k}\right) - \nabla_x g\left(w_{t,k}, z_{t,k}\right)\right)$

7:     $x_{t,k+1} \leftarrow w_{t,k} - \frac{1}{L}\hat{\nabla} L_\lambda^*(w_{t,k})$

8:     $w_{t,k+1} \leftarrow x_{t,k+1} + \theta_{k+1}\left(x_{t,k+1} - x_{t,k}\right)$

9:     $k \leftarrow k + 1, K \leftarrow K + 1$

10:     **if** $(k+1)^{4+\nu_f} H_\nu^2 S_k^{\nu_f} > L^2$ **then**

11:        $x_{t+1,0} \leftarrow x_{t,k}$

12:        $y_{t+1,-1} \leftarrow 0, z_{t+1,-1} \leftarrow 0, w_{t+1,0} \leftarrow x_{t+1,0}$

13:        $k \leftarrow 0, t \leftarrow t + 1$

14:     **end if**

15: **until** $\|\nabla L_\lambda(\bar{w}_{t,k})\| \le \epsilon$

16: **Output:** averaged solution $\bar{w}_{t,k}$ defined by (7)

---

**Averaged Solution.** Inspired by [23], we set $\theta_k = \frac{k}{k+1}$ and define

$$\bar{w}_k = \sum_{i=0}^{k-1} p_{k,i} w_i, \tag{7}$$

where $p_{k,i} = \frac{2(i+1)}{k(k+1)}$. We can update $\bar{w}_k$ in the following manner: $\bar{w}_k = \frac{k-1}{k+1}\bar{w}_{k-1} + \frac{2}{k+1}w_{k-1}$.

The following lemma shows that $\{x_i\}_{i=0}^{k-1}$ and $\{w_i\}_{i=0}^{k-1}$ are bounded within any epoch ending at iteration $k$.

**Lemma 3.** *Let Assumption 1 holds, $H_\nu$ and $\mathcal{D} = \mathcal{R}$ be given in equation 6, and $\bar{w}_k$ be defined in equation 7. For any epoch ending at iteration $k$, the following holds:*

$$\max_{0 \le i \le j \le k-1} \|x_i - x_j\| \le \mathcal{D}, \quad \max_{0 \le i \le k-1} \|w_i - \bar{w}_k\| \le \max_{0 \le i \le j \le k-1} \|w_i - w_j\| \le \mathcal{D}.$$

**Condition 1** (Inexact gradients). *Under Assumption 1 and given $\sigma > 0$, we assume that the estimators $y_{t,i}$ and $z_{t,i}$ satisfy the conditions*

$$\|z_{t,i} - y^*(w_{t,i})\| \le \frac{\sigma}{2\lambda L_g}, \quad \|y_{t,i} - y_\lambda^*(w_{t,i})\| \le \frac{\sigma}{4\lambda L_g}, \tag{8}$$

*for any $t$-th epoch ending at iteration $k$, where $i = 0, \ldots, k-1$.*

**Remark 1.** *It is noteworthy that Condition 1 holds in Algorithm 1 as long as the inner loop iteration number $T_{t,k}$ and $T'_{t,k}$ are large enough. This will be formally addressed in our convergence analysis later, in Theorem 2.*

Under Condition 1, the bias of $\nabla L_\lambda^*(w_{t,k})$ and its estimator $\hat{\nabla} L_\lambda^*(w_{t,k})$ can be bounded as shown below:

**Lemma 4** (Inexact gradients). *Under Assumption 1 and supposing that Condition 1 holds, we have*

$$\|\nabla L_\lambda^*(w_{t,i}) - \hat{\nabla} L_\lambda^*(w_{t,i})\| \le \sigma$$

*for any $t$-th epoch ending at iteration $k$, where $i = 0, \ldots, k-1$.*

## 4 COMPLEXITY ANALYSIS

In this section, we analyze the performance of Algorithm 1. We begin in Section 4.1 by presenting several useful lemmas that rely on the boundedness of the iterates generated within a single epoch.

These results serve as key tools for our subsequent analysis. We then establish the descent property of the objective function and derive an upper bound for $\|\nabla L_\lambda^*(\bar{w}_i)\|$. Finally, in Section 4.2, we present the main complexity results for Algorithm 1.

## 4.1 TOOLS FOR ANALYSIS

We use the following two Hessian-free inequalities to analyze the complexity of Algorithm 1.

**Lemma 5.** *Under Assumption 1 and with $\lambda \geq 2L_f/\mu$, the following holds for any $x_1, \ldots, x_n$ satisfying $\max_{1 \leq i \leq j \leq n} \|x_i - x_j\| \leq \mathcal{D}$ and $q_1, \ldots, q_n \geq 0$ such that $\sum_{q=1}^n q_i = 1$:*

$$\left\| \nabla L_\lambda^*(\sum_{i=1}^n q_i x_i) - \sum_{i=1}^n q_i \nabla L_\lambda^*(x_i) \right\| \leq \frac{H_\nu}{1 + \nu_f} \left( \sum_{1 \leq i < j \leq n} q_i q_j \|x_i - x_j\|^2 \right)^{\frac{1 + \nu_f}{2}},$$

*where $H_\nu$ and $\mathcal{D}$ are defined in (6).*

**Lemma 6.** *Under Assumption 1 and with $\lambda \geq 2L_f/\mu$, the following holds for any $x$ and $x'$ satisfying $\|x - x'\| \leq \mathcal{D}$:*

$$L_\lambda^*(x) - L_\lambda^*(x') \leq \frac{1}{2}\langle \nabla L_\lambda^*(x) + \nabla L_\lambda^*(x'), x - x' \rangle + \frac{2H_\nu}{(1 + \nu_f)(2 + \nu_f)(3 + \nu_f)} \|x - x'\|^{2 + \nu_f},$$

*where $H_\nu$ and $\mathcal{D}$ are defined in (6).*

Lemma 5 bounds the discrepancy between the average gradient over an epoch and the true gradient at the averaged iterate $\bar{w}_k$ defined in (7), while Lemma 6 establishes a quadratic surrogate inequality for the function difference, which serves as a key ingredient for showing descent of the potential function. In light of these lemmas and following [23], we define the potential function $\Phi_k$ as

$$\Phi_k := L_\lambda^*(x_k) + \frac{\theta_k^2}{2} \left( \frac{1}{2L} \|\nabla L_\lambda^*(x_{k-1}) + L(x_k - x_{k-1})\|^2 + \frac{L}{2} \|x_k - x_{k-1}\|^2 \right). \quad (9)$$

The following lemma shows that $\Phi_k$ is a decreasing sequence if $\|x_k - x_{k-1}\|$ and $\sigma$ are sufficiently small.

**Lemma 7.** *Suppose that Assumption 1, Condition 1, and $\lambda \geq 2L_f/\mu$ hold. Then we have*

$$\Phi_{k+1} - \Phi_k \leq \|x_k - x_{k-1}\|^{2 + \nu_f} \left( \frac{2H_\nu}{(1 + \nu_f)(2 + \nu_f)(3 + \nu_f)} \theta_k^{2 + \nu_f} + \frac{H_\nu}{1 + \nu_f} \theta_k^{\frac{3 + \nu_f}{2}} \right)$$

$$+ \|x_k - x_{k-1}\|^{2 + 2\nu_f} \frac{2H_\nu^2}{(1 + \nu_f)^2} \frac{\theta_k^{2 + \nu_f}}{L} + \frac{\theta_{k+1}^2 + \theta_k - 2}{4} L \|x_{k+1} - x_k\|^2$$

$$- \frac{\theta_k^2}{4L} \|\nabla L_\lambda^*(x_k)\|^2 + \frac{\sigma^2}{2L} + \sigma \|x_{k+1} - x_k\|. \quad (10)$$

Moreover, we can leverage this potential decrease to quantify the reduction of $L_\lambda^*(\cdot)$ over an entire epoch. The following lemma shows that $L_\lambda^*(x)$ decreases whenever $S_k > 0$ and $\sigma$ is sufficiently small.

**Lemma 8.** *Suppose that Assumption 1, Condition 1, and $\lambda \geq 2L_f/\mu$ hold. Then the decrease value of $L_\lambda^*(\cdot)$ in one epoch satisfies:*

$$L_\lambda^*(x_k) - L_\lambda^*(x_0) \leq -\frac{LS_k}{32k} + \frac{k\sigma^2}{2L} + \sigma \sum_{i=0}^{k-1} \|x_{i+1} - x_i\|. \quad (11)$$

The following lemma provides an upper bound on the minimum gradient norm of the penalized objective $L_\lambda^*$ evaluated at the averaged iterates $\{\bar{w}_i\}_{i=1}^{k-1}$.

**Lemma 9.** *Suppose that Assumption 1, Condition 1, and $\lambda \geq 2L_f/\mu$ hold. The following is true when $k \geq 2$:*

$$\min_{1 \leq i < k} \|\nabla L_\lambda^*(\bar{w}_i)\| \leq \sigma + cL\sqrt{S_{k-1}/k^3},$$

*where $c = 2\sqrt{6} + 27$.*

## 4.2 MAIN RESULTS

In the following proposition, we show that the iteration complexity of the outer loop is bounded.

**Proposition 1.** *Suppose that Assumption 1, Condition 1, and $\lambda \geq 2L_f/\mu$ hold. Let $c = 2\sqrt{6} + 27$ as defined in Lemma 9, and define $\Delta_\lambda = L_\lambda^*(x_{0,0}) - \min_{x \in \mathbb{R}^{d_x}} L_\lambda^*(x)$. Let*

$$(\alpha, \beta) = (\frac{1}{L_g}, \frac{\sqrt{L_g} - \sqrt{\mu}}{\sqrt{L_g} + \sqrt{\mu}}), \quad (\alpha', \beta') = (\frac{1}{2\lambda L_g}, \frac{\sqrt{4L_g} - \sqrt{\mu}}{\sqrt{4L_g} + \sqrt{\mu}}),$$

$$\theta_k = \frac{k}{k+1} \quad and \quad \sigma = \frac{1}{64c+1}\epsilon. \tag{12}$$

*Algorithm 1 terminates within*

$$\mathcal{O}\left(\Delta_\lambda \lambda^{\frac{\nu_f(1-\nu_g)}{(2+2\nu_f)}} \ell^{\frac{2+\nu_f}{2+2\nu_f}} \kappa^{\frac{6+4\nu_f+\nu_f\nu_g}{(2+2\nu_f)}} \epsilon^{-\frac{4+3\nu_f}{2+2\nu_f}}\right)$$

*total iterations, outputting $\bar{w}_{t,k}$ satisfying $\|\nabla L_\lambda^*(\bar{w}_{t,k})\| \leq \epsilon$. Moreover, Algorithm 1 terminates within*

$$\mathcal{O}\left(\Delta_\lambda \lambda^{\frac{1-\nu_g}{(2-\nu_f)(1+\nu_f)}} \ell^{\frac{1}{1+\nu_f}} \kappa^{\frac{8-3\nu_f}{(2-\nu_f)(1+\nu_f)}} \epsilon^{-\frac{2+\nu_f}{2+2\nu_f}}\right)$$

*epochs.*

We present the complexity analysis of our algorithm, aiming to establish its guarantee for finding an $\mathcal{O}(\epsilon)$-stationary point of problem (1).

**Theorem 1.** *Suppose that both Assumption 1 and Condition 1 hold. Define $\Delta = \varphi(x_{0,0}) - \min_{x \in \mathbb{R}^{d_x}} \varphi(x)$. Let $\lambda = \max(\mathcal{O}(\kappa), \mathcal{O}(\ell\kappa^3)/\epsilon, \mathcal{O}(\ell\kappa^2)/\Delta)$ and set the other parameters as specified in equation 12, Algorithm 1 terminates within*

$$\mathcal{O}\left(\Delta \ell^{\frac{2+2\nu_f - \nu_f\nu_g}{2+2\nu_f}} \kappa^{\frac{6+7\nu_f-2\nu_f\nu_g}{2+2\nu_f}} \epsilon^{-\frac{4+4\nu_f-\nu_f\nu_g}{2+2\nu_f}}\right)$$

*iterates, outputting $\bar{w}_{t,k}$ satisfying $\|\nabla\varphi(\bar{w}_k)\| \leq 2\epsilon$. Moreover, Algorithm 1 terminates within*

$$\mathcal{O}\left(\Delta \ell^{\frac{1+\nu_f - \nu_f\nu_g}{1+\nu_f}} \kappa^{\frac{3+4\nu_f-2\nu_f\nu_g}{1+\nu_f}} \epsilon^{-\frac{2+2\nu_f-\nu_f\nu_g}{1+\nu_f}}\right)$$

*epochs.*

When $\nu_f = \nu_g = 1$, Theorem 1 shows that within $\mathcal{O}\left(\Delta\ell^{3/4}\kappa^{11/4}\epsilon^{-7/4}\right)$ outer iterations and $\mathcal{O}(\Delta\ell^{1/2}\kappa^{5/2}\epsilon^{-3/2})$ epochs, the algorithm will find an $\mathcal{O}(\epsilon)$-stationary point.

**Remark 2.** *Throughout the proof, we only use the restricted Hölder and Lipschitz properties, where restricted Lipschitz continuity can be defined analogously to Definition 1. Therefore, the assumption on global Lipschitz and Hölder smoothness in Assumption 1 can be relaxed to restricted smoothness.*

To make Condition 1 hold, it suffices to run AGD for a sufficiently large number of iterations, which only introduces a logarithmic factor to the total complexity. This gives the following result.

**Theorem 2.** *Suppose that Assumption 1 holds. In the $t$-th epoch, we set the inner-loop iteration numbers $T_{t,k}$ and $T'_{t,k}$ according to equation 44, equation 45, equation 46, and equation 47 in Appendix E. We then run Algorithm 1 with the parameters specified in Theorem 1. Under these settings, all $y_{t,k}$ and $z_{t,k}$ satisfy Condition 1. Moreover, the total first-order oracle complexity is*

$$\tilde{\mathcal{O}}\left(\Delta \ell^{\frac{2+2\nu_f - \nu_f\nu_g}{2+2\nu_f}} \kappa^{\frac{7+8\nu_f-2\nu_f\nu_g}{2+2\nu_f}} \epsilon^{-\frac{4+4\nu_f-\nu_f\nu_g}{2+2\nu_f}}\right).$$

When $\nu_f = \nu_g = 1$, the first-order oracle complexity is $\tilde{\mathcal{O}}\left(\Delta\ell^{3/4}\kappa^{13/4}\epsilon^{-7/4}\right)$. This matches the $\tilde{\mathcal{O}}(\epsilon^{-7/4})$ rate obtained independently and concurrently by [13], and also improves upon the earlier result of [15], as shown in Table 1. We defer the proof to Appendix E. Under the Hölder continuity assumption, to the best of our knowledge, we are the first to propose a method that finds an $\epsilon$-stationary point. Furthermore, under the Lipschitz continuity assumption, our approach outperforms all existing methods in the literature, as the proposed method RAGD-GS relies solely on first-order oracle information, which is in line with the concurrent work [13].

## 5 NUMERICAL EXPERIMENT

This section compares the performance of the proposed method with several existing methods, including RAHGD [15], BA [29], AID [26], ITD [26], F²BA [13] and AccF²BA [13]. For the bilevel approximation (BA) method introduced in [29], we implement a conjugate gradient approach to compute Hessian-vector products since the original work doesn't specify this computational detail. We refer to this modified version as BA-CG to distinguish it from other algorithm. To quantify variability, each experiment is repeated over 5 independent trials, and we report the average performance. Our experiments were conducted on a PC with Intel Core i7-13650HX CPU (2.60GHz, 20 cores), 24GB RAM, and the platform is 64-bit Windows 11 Home Edition (version 26100).

### 5.1 DATA HYPERCLEANING

Data hypercleaning ([30]; [28]) is a bilevel optimization problem aimed at cleaning noisy labels in datasets. The cleaned data forms the validation set, while the rest serves as the training set. The problem is formulated as:

$$\min_{\lambda \in \mathbb{R}^{N_{\text{tr}}}} \ f(W^*(\lambda), \lambda) = \frac{1}{|\mathcal{D}_{\text{val}}|} \sum_{(x_i, y_i) \in \mathcal{D}_{\text{val}}} -\log(y_i^\top W^*(\lambda) x_i)$$

$$\text{s.t. } W^*(\lambda) = \arg\min_{W \in \mathbb{R}^{d_y \times d_x}} \frac{1}{|\mathcal{D}_{\text{tr}}|} \sum_{(x_i, y_i) \in \mathcal{D}_{\text{tr}}} -\sigma(\lambda_i) \log(y_i^\top W x_i) + C_r \|W\|^2,$$

where $\mathcal{D}_{\text{tr}}$ and $\mathcal{D}_{\text{val}}$ are the training and validation sets, respectively, $W$ is the weight matrix of the classifier, $\sigma(\cdot)$ is the sigmoid function, and $C_r$ is a regularization parameter. In our experiments, we follow [30] and set $C_r = 0.001$.

For MNIST [31], we use $|\mathcal{D}_{\text{tr}}| = 20{,}000$ training samples (partially noisy) and $|\mathcal{D}_{\text{val}}| = 5{,}000$ clean validation samples, with corruption rate $p$ indicating the ratio of noisy labels in the training set. In Figures 1 and 2, inner and outer learning rates are searched over $\{0.001, 0.01, 0.1, 1, 10, 100\}$. For all methods except BA, inner GD/AGD steps are from $\{50, 100, 200, 500\}$; for BA, we choose GD steps from $\{\lceil c(k+1)^{1/4} \rceil : c \in \{0.5, 1, 2, 4\}\}$ as in [29]. For F²BA, AccF²BA and our method, $\lambda$ is selected from $\{100, 300, 500, 700\}$. The results, shown in Figures 1 and 2, demonstrate that our proposed method achieves acceleration effects comparable to those in [13; 15], and outperforms all other methods.

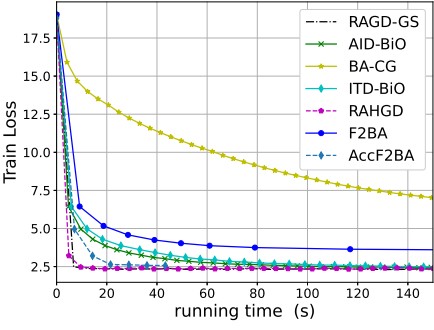

Figure 1: Corruption rate $p = 0.2$

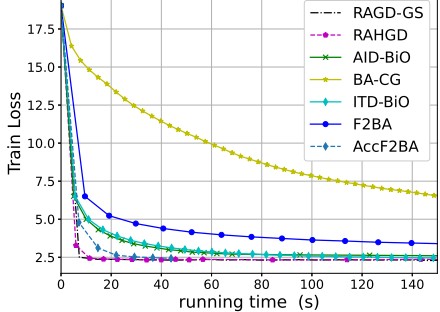

Figure 2: Corruption rate $p = 0.4$

### 5.2 HYPERPARAMETER OPTIMIZATION

Hyperparameter optimization is a bilevel optimization task aimed at minimizing the validation loss. We compare our proposed algorithms with baseline algorithms on the 20 Newsgroups dataset [11], which consists of 18,846 news articles divided into 20 topics, with 130,170 sparse tf-idf features. The dataset is split into training, validation, and test sets with sizes $|\mathcal{D}_{\text{tr}}| = 5{,}657$, $|\mathcal{D}_{\text{val}}| = 5{,}657$,

and $|\mathcal{D}_{\text{test}}| = 7{,}532$, respectively. The optimization problem is formulated as:

$$\min_{\lambda \in \mathbb{R}^p} \frac{1}{|\mathcal{D}_{\text{val}}|} \sum_{(x_i, y_i) \in \mathcal{D}_{\text{val}}} L(w^*(\lambda); x_i, y_i)$$

$$\text{s.t.} \quad w^*(\lambda) = \arg\min_{w \in \mathbb{R}^{c \times p}} \frac{1}{|\mathcal{D}_{\text{tr}}|} \sum_{(x_i, y_i) \in \mathcal{D}_{\text{tr}}} L(w; x_i, y_i) + \frac{1}{2cp} \sum_{j=1}^{c} \sum_{k=1}^{p} \exp(\lambda_k) w_{jk}^2.$$

For the evaluation in Figure 3, inner and outer learning rates are selected from {0.001, 0.01, 0.1, 1, 10, 100}, and GD/AGD steps from {5, 10, 30, 50}. For BA, we choose GD steps from $\{\lceil c(k+1)^{1/4} \rceil : c \in \{0.5, 1, 2, 4\}\}$ as in [29]. For F²BA, AccF²BA and our method, $\lambda$ is chosen from {100, 300, 500, 700}. As shown in Figure 3, our proposed method exhibits performance comparable to that of [13; 15], while significantly outperforming other competing algorithms by converging faster and reaching a lower test loss.

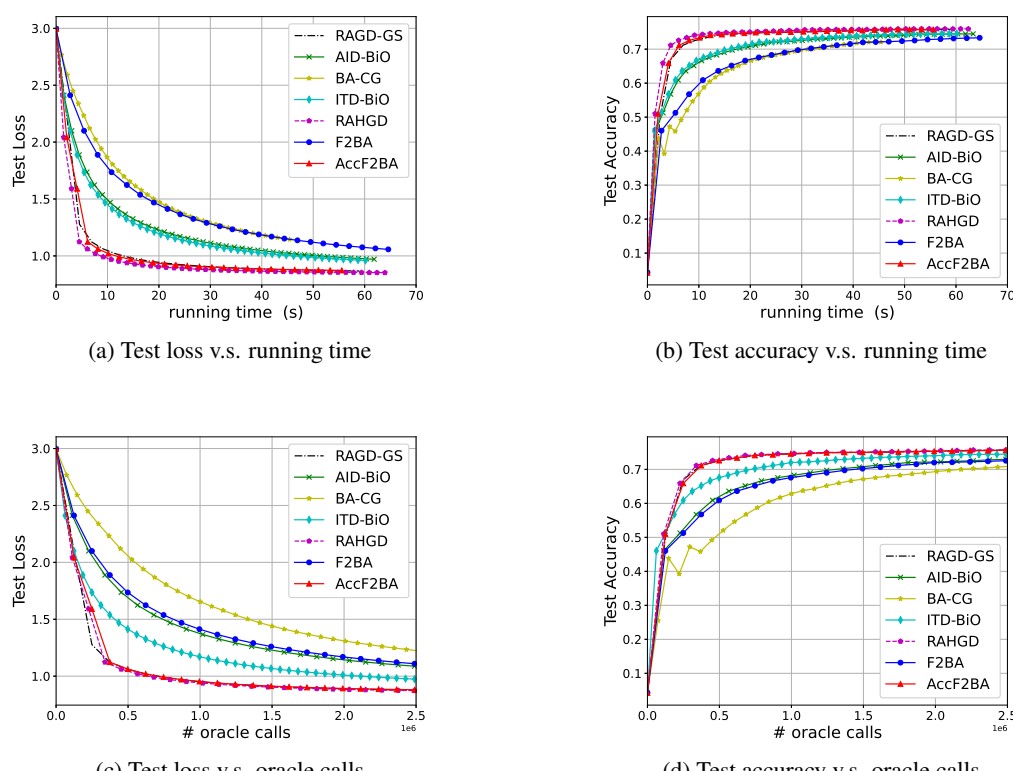

(a) Test loss v.s. running time      (b) Test accuracy v.s. running time

(c) Test loss v.s. oracle calls      (d) Test accuracy v.s. oracle calls

Figure 3: Results of test loss and test accuracy evaluated on the test set.

# 6 CONCLUSION

This work introduces an accelerated first-order framework for solving nonconvex–strongly convex bilevel optimization problems, extending nonconvex optimization techniques to a broader setting under generalized Hölder continuity. With a carefully designed restart condition, the iterates remain uniformly bounded within each epoch, ensuring stability and convergence. We further provide oracle complexity bounds with rigorous error analysis and convergence guarantees. Our theory is supported by empirical evidence, demonstrating the effectiveness and robustness of the algorithm. While recent advances in the stochastic setting [14; 32; 13] mainly focus on the first-order oracle complexity, it remains unclear whether acceleration with an appropriate restart scheme is attainable under higher-order smoothness assumptions ($\nabla^2 f$ and $\nabla^3 g$). Challenges such as noisy restart triggers and precise hyper-gradient estimation make this nontrivial. We leave this challenging direction for future work.

ETHICS STATEMENT

This work does not present any apparent ethical concerns. The proposed algorithms are purely theoretical and experimental in nature, and they do not involve human subjects, sensitive personal data, or applications that pose foreseeable risks of harm. Nevertheless, we recognize the importance of ethical considerations in machine learning research and adhere to the ICLR Code of Ethics.

REPRODUCIBILITY STATEMENT

To ensure reproducibility, we provide the following: (1) all theoretical results are accompanied by complete proofs in the appendix; (2) experimental setups, including dataset preprocessing and hyperparameter settings, are described in detail; (3) source code implementing our algorithms will be made available in the supplementary material. These resources should allow others to fully replicate our findings.

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

APPENDIX

This appendix provides additional theoretical results and technical proofs that support the main text. For clarity, we organize the appendix to follow the structure of the main paper: each subsection presents the detailed derivations and omitted proofs of the corresponding lemmas and theorems.

## A   THE USE OF LARGE LANGUAGE MODELS (LLMS)

No large language models (LLMs) were used in the development of the research ideas, theoretical results, experiments, or writing of this paper. All contents are solely the work of the authors.

## B   NOTATIONS FOR TENSORS

We adopt the tensor notation from [33]. For a three-way tensor $\mathcal{X} \in \mathbb{R}^{d_1 \times d_2 \times d_3}$, the entry at $(i_1, i_2, i_3)$ is denoted by $[\mathcal{X}]_{i_1, i_2, i_3}$. The inner product between $\mathcal{X}$ and $\mathcal{Y}$ is defined as

$$\langle \mathcal{X}, \mathcal{Y} \rangle := \sum_{i_1, i_2, i_3} [\mathcal{X}]_{i_1, i_2, i_3} [\mathcal{Y}]_{i_1, i_2, i_3}.$$

The operator norm is

$$\|\mathcal{X}\| := \sup_{\|x_1\| = \|x_2\| = \|x_3\| = 1} \langle \mathcal{X}, x_1 \circ x_2 \circ x_3 \rangle,$$

where $[x_1 \circ x_2 \circ x_3]_{i_1, i_2, i_3} := [x_1]_{i_1} [x_2]_{i_2} [x_3]_{i_3}$. This definition generalizes the matrix spectral norm and the Euclidean norm for vectors to three-way tensors. Let $\mathcal{X} \in \mathbb{R}^{d_1 \times d_2 \times d_3}$ be a three-way tensor, and let $A \in \mathbb{R}^{d_1' \times d_1}$ be a matrix. The mode-1 product of $\mathcal{X}$ and $A$, denoted by $\mathcal{X} \times_1 A \in \mathbb{R}^{d_1' \times d_2 \times d_3}$, is defined component-wise as

$$[\mathcal{X} \times_1 A]_{i_1', i_2, i_3} := \sum_{i_1=1}^{d_1} A_{i_1', i_1} \mathcal{X}_{i_1, i_2, i_3}.$$

Mode-2 and mode-3 products, denoted by $\mathcal{X} \times_2 B$ and $\mathcal{X} \times_3 C$, are defined analogously for matrices $B \in \mathbb{R}^{d_2' \times d_2}$ and $C \in \mathbb{R}^{d_3' \times d_3}$, respectively. Moreover, the operator norm satisfies the submultiplicative property under mode-$i$ multiplication:

$$\|\mathcal{X} \times_i A\| \le \|A\| \cdot \|\mathcal{X}\|, \quad \text{for } i = 1, 2, 3.$$

## C   PROOF OF LEMMAS IN SECTION 2

**Lemma C.1** (Lemma B.2 by [13]). *Under Assumption 1, for $\lambda \ge 2L_f / \mu$, it holds that $\|y_\lambda^\star(x) - y^\star(x)\| \le \frac{C_f}{\lambda \mu}$.*

**Lemma C.2** (Lemma B.5 by [13]). *Under Assumption 1, for $\lambda \ge 2L_f / \mu$, it holds that $\|\nabla y^\star(x) - \nabla y_\lambda^\star(x)\| \le D_2 / \lambda$, where*

$$D_2 := \left( \frac{1}{\mu} + \frac{2L_g}{\mu^2} \right) \left( L_f + \frac{C_f \rho_g}{\mu} \right) = \mathcal{O}\left( \kappa^3 \right).$$

**Lemma C.3** (Lemma B.6 by [13]). *Under Assumption 1, for $\lambda \ge 2L_f / \mu$, it holds that $\|\nabla y^\star(x)\| \le L_g / \mu$, $\|\nabla y_\lambda^\star(x)\| \le 4L_g / \mu$.*

This implies that $y^\star(x)$ is $(L_g / \mu)$-Lipschitz continuous, $y_\lambda^\star(x)$ is $(4L_g / \mu)$-Lipschitz continuous.

**Lemma C.4.** *Under Assumption 1, for $\lambda \ge 2L_f / \mu$, we have*

$$\|\nabla^2 y^\star(x) - \nabla^2 y_\lambda^\star(x)\| \le \frac{D_4}{\lambda^{\nu_g}},$$

*where*

$$D_4 := \frac{2\rho_g}{\mu^2}(\frac{\mu}{2L_f})^{1-\nu_g}\left(1+\frac{L_g}{\mu}\right)^2\left(L_f + \frac{C_f\rho_g}{\mu}\right) + \frac{14L_g\rho_g D_2}{\mu^2}(\frac{\mu}{2L_f})^{1-\nu_g}$$
$$+ \frac{50L_g^2}{\mu^3}\left(\rho_f(\frac{\mu}{2L_f})^{1-\nu_g} + M_g(\frac{C_f}{\mu})^{\nu_g}\right)$$
$$= \mathcal{O}(\kappa^{4+\nu_g}).$$

*Proof.* We begin by differentiating the identity

$$\nabla_{xy}^2 g\left(x, y^*(x)\right) + \nabla y^*(x)\nabla_{yy}^2 g\left(x, y^*(x)\right) = 0$$

with respect to $x$. This yields

$$\nabla_{xxy}^3 g\left(x, y^*(x)\right) + \nabla_{yxy}^3 g\left(x, y^*(x)\right) \times_1 \nabla y^*(x) + \nabla^2 y^*(x) \times_3 \nabla_{yy}^2 g\left(x, y^*(x)\right)$$
$$+ \nabla_{xyy}^3 g\left(x, y^*(x)\right) \times_2 \nabla y^*(x) + \nabla_{yyy}^3 g\left(x, y^*(x)\right) \times_1 \nabla y^*(x) \times_2 \nabla y^*(x) = 0.$$

Rearranging terms to isolate $\nabla^2 y^*(x)$, we obtain

$$\nabla^2 y^*(x)$$
$$= -\left(\nabla_{xxy}^3 g\left(x, y^*(x)\right) + \nabla_{yxy}^3 g\left(x, y^*(x)\right) \times_1 \nabla y^*(x)\right) \times_3 \left[\nabla_{yy}^2 g\left(x, y^*(x)\right)\right]^{-1}$$
$$- \nabla_{xyy}^3 g\left(x, y^*(x)\right) \times_2 \nabla y^*(x) \times_3 \left[\nabla_{yy}^2 g\left(x, y^*(x)\right)\right]^{-1}$$
$$- \nabla_{yyy}^3 g\left(x, y^*(x)\right) \times_1 \nabla y^*(x) \times_2 \nabla y^*(x) \times_3 \left[\nabla_{yy}^2 g\left(x, y^*(x)\right)\right]^{-1}. \tag{13}$$

Analogously, we have

$$\nabla^2 y_\lambda^*(x)$$
$$= -\left(\nabla_{xxy}^3 L_\lambda\left(x, y_\lambda^*(x)\right) + \nabla_{yxy}^3 L_\lambda\left(x, y_\lambda^*(x)\right) \times_1 \nabla y_\lambda^*(x)\right) \times_3 \left[\nabla_{yy}^2 L_\lambda\left(x, y_\lambda^*(x)\right)\right]^{-1}$$
$$- \nabla_{xyy}^3 L_\lambda\left(x, y_\lambda^*(x)\right) \times_2 \nabla y_\lambda^*(x) \times_3 \left[\nabla_{yy}^2 L_\lambda\left(x, y_\lambda^*(x)\right)\right]^{-1}$$
$$- \nabla_{yyy}^3 L_\lambda\left(x, y_\lambda^*(x)\right) \times_1 \nabla y_\lambda^*(x) \times_2 \nabla y_\lambda^*(x) \times_3 \left[\nabla_{yy}^2 L_\lambda\left(x, y_\lambda^*(x)\right)\right]^{-1}. \tag{14}$$

Next, we estimate the difference between the corresponding third-order derivatives in the original and penalized problems. To begin with, we observe that

$$\left\|\nabla_{xxy}^3 g\left(x, y^*(x)\right) - \frac{\nabla_{xxy}^3 L_\lambda\left(x, y_\lambda^*(x)\right)}{\lambda}\right\| \le M_g\left\|y_\lambda^*(x) - y^*(x)\right\|^{\nu_g} + \frac{\rho_f}{\lambda} = \frac{\rho_f}{\lambda} + M_g\left(\frac{C_f}{\lambda\mu}\right)^{\nu_g}.$$

Similarly, for the mixed partial derivative and its contraction with $\nabla y^*(x)$, we have

$$\left\|\nabla_{yxy}^3 g\left(x, y^*(x)\right) \times_1 \nabla y^*(x) - \frac{\nabla_{yxy}^3 L_\lambda\left(x, y_\lambda^*(x)\right) \times_1 \nabla y_\lambda^*(x)}{\lambda}\right\|$$

$$\le \left\|\nabla y^*(x) - \nabla y_\lambda^*(x)\right\|\left\|\nabla_{yxy}^3 g\left(x, y^*(x)\right)\right\| + \left\|\nabla y_\lambda^*(x)\right\|\left\|\nabla_{yxy}^3 g\left(x, y^*(x)\right) - \frac{\nabla_{yxy}^3 L_\lambda\left(x, y_\lambda^*(x)\right)}{\lambda}\right\|$$

$$\le \frac{\rho_g D_2}{\lambda} + \frac{4L_g}{\mu}\left(\frac{\rho_f}{\lambda} + M_g\left(\frac{C_f}{\lambda\mu}\right)^{\nu_g}\right).$$

Furthermore, we control the error in the third-order term involving two contractions:

$$\left\| \nabla^3_{yyy} g\left(x, y^*(x)\right) \times_1 \nabla y^*(x) \times_2 \nabla y^*(x) - \frac{\nabla^3_{yyy} L_\lambda\left(x, y^*_\lambda(x)\right) \times_1 \nabla y^*_\lambda(x) \times_2 \nabla y^*_\lambda(x)}{\lambda} \right\|$$

$$\leq \|\nabla y^*(x)\| \left\| \nabla^3_{yyy} g\left(x, y^*(x)\right)\right\| \|\nabla y^*(x) - \nabla y^*_\lambda(x)\|$$

$$+ \|\nabla y^*_\lambda(x)\| \left\| \nabla^3_{yyy} g\left(x, y^*(x)\right)\right\| \|\nabla y^*(x) - \nabla y^*_\lambda(x)\|$$

$$+ \|\nabla y^*_\lambda(x)\|^2 \left\| \nabla^3_{xxy} g\left(x, y^*(x)\right) - \frac{\nabla^3_{xxy} L_\lambda\left(x, y^*_\lambda(x)\right)}{\lambda} \right\|$$

$$\leq \frac{5 L_g \rho_g D_2}{\lambda \mu} + \frac{16 L_g^2}{\mu^2}\left(\frac{\rho_f}{\lambda} + M_g \left(\frac{C_f}{\lambda \mu}\right)^{\nu_g}\right).$$

Combining the above inequalities, we are now ready to bound the difference between the second derivatives:

$$\left\| \nabla^2 y^*(x) - \nabla^2 y^*_\lambda(x)\right\|$$

$$\leq \rho_g \left(1 + \frac{L_g}{\mu}\right)^2 \left\| \left[\nabla^2_{yy} g\left(x, y^*(x)\right)\right]^{-1} - \left[\frac{\nabla^2_{yy} L_\lambda\left(x, y^*_\lambda(x)\right)}{\lambda}\right]^{-1}\right\|$$

$$+ \left(\frac{7 L_g \rho_g D_2}{\lambda \mu} + \frac{25 L_g^2}{\mu^2}\left(\frac{\rho_f}{\lambda} + M_g\left(\frac{C_f}{\lambda\mu}\right)^{\nu_g}\right)\right)\left\| \left[\frac{\nabla^2_{yy} L_\lambda\left(x, y^*_\lambda(x)\right)}{\lambda}\right]^{-1}\right\|$$

$$\leq \frac{2\rho_g}{\lambda\mu^2}\left(1 + \frac{L_g}{\mu}\right)^2 \left(L_f + \frac{C_f \rho_g}{\mu}\right) + \frac{14 L_g \rho_g D_2}{\lambda\mu^2} + \frac{50 L_g^2}{\mu^3}\left(\frac{\rho_f}{\lambda} + M_g\left(\frac{C_f}{\lambda\mu}\right)^{\nu_g}\right)$$

$$\leq \frac{D_4}{\lambda^{\nu_g}}.$$

$$\square$$

**Lemma C.5.** *Under Assumption 1, for $\lambda \geq 2L_f/\mu$, the mappings $\nabla y^*(x)$ and $\nabla y^*_\lambda(x)$ are Lipschitz continuous with constants $\left(1 + \frac{L_g}{\mu}\right)^2 \frac{\rho_g}{\mu}$ and $\left(1 + \frac{4L_g}{\mu}\right)^2 \left(\frac{2\rho_g}{\mu} + \frac{\rho_f}{L_f}\right)$, respectively.*

*Proof.* Recall that

$$\nabla y^*_\lambda(x) = -\nabla^2_{xy} L_\lambda\left(x, y^*_\lambda(x)\right) \left[\nabla^2_{yy} L_\lambda\left(x, y^*_\lambda(x)\right)\right]^{-1},$$

and

$$\nabla y^*(x) = -\nabla^2_{xy} g\left(x, y^*(x)\right) \left[\nabla^2_{yy} g\left(x, y^*(x)\right)\right]^{-1}.$$

By equation 13 and equation 14, we can obtain the Lipschitz constants of $\nabla y^*(x)$ and $\nabla y^*_\lambda(x)$ by directly bounding $\|\nabla^2 y^*(x)\|$ and $\|\nabla^2 y^*_\lambda(x)\|$. Specifically, we have

$$\|\nabla^2 y^*(x)\| \leq \frac{1}{\mu}\left(\rho_g + \rho_g \frac{L_g}{\mu} + \rho_g \frac{L_g}{\mu} + \rho_g \left(\frac{L_g}{\mu}\right)^2\right) = \frac{\rho_g}{\mu}\left(1 + \frac{L_g}{\mu}\right)^2,$$

$$\|\nabla^2 y^*_\lambda(x)\| \leq \frac{2}{\lambda\mu}(\rho_f + \lambda\rho_g)\left(1 + 2\frac{4L_g}{\mu} + \left(\frac{4L_g}{\mu}\right)^2\right) \leq \left(1 + \frac{4L_g}{\mu}\right)^2\left(\frac{2\rho_g}{\mu} + \frac{\rho_f}{L_f}\right).$$

Here we use Lemma C.3, $\lambda \geq 2L_f/\mu$, $\|\nabla^3_{xxy} g(x,y)\| \leq \rho_g$, $\|\nabla^3_{xyy} g(x,y)\| \leq \rho_g$, $\|\nabla^3_{yyy} g(x,y)\| \leq \rho_g$, $\|\nabla^2_{yy} g(x,y)\| \geq \mu$, $\|\nabla^2_{yy} L_\lambda(x,y)\| \geq \frac{1}{2}\lambda\mu$, $\|\nabla^3_{xxy} f(x,y)\| \leq \rho_f$, $\|\nabla^3_{xyy} f(x,y)\| \leq \rho_f$ and $\|\nabla^3_{yyy} f(x,y)\| \leq \rho_f$.

$$\square$$

## C.1 PROOF OF LEMMA 2

*Proof.* We decompose $\nabla^2 L_\lambda^*(x)$ into two components:

$$\nabla^2 L_\lambda^*(x) = A(x) + B(x),$$

where

$$A(x) = \nabla^2_{xx} f\left(x, y_\lambda^*(x)\right) + \nabla y_\lambda^*(x) \nabla^2_{yx} f\left(x, y_\lambda^*(x)\right)$$

and

$$B(x) = \lambda \left(\nabla^2_{xx} g\left(x, y_\lambda^*(x)\right) - \nabla^2_{xx} g\left(x, y^*(x)\right)\right)$$
$$+ \lambda \left(\nabla y_\lambda^*(x) \nabla^2_{yx} g\left(x, y_\lambda^*(x)\right) - \nabla y^*(x) \nabla^2_{yx} g\left(x, y^*(x)\right)\right).$$

To analyze the variation of $A(x)$, we observe:

$$\begin{aligned}
&\|A(x_1) - A(x_2)\| \\
\leq & \|\nabla^2_{xx} f\left(x_1, y_\lambda^*(x_1)\right) - \nabla^2_{xx} f\left(x_2, y_\lambda^*(x_2)\right)\| \\
& + \|\nabla y_\lambda^*(x_1) \nabla^2_{yx} f\left(x_1, y_\lambda^*(x_1)\right) - \nabla y_\lambda^*(x_2) \nabla^2_{yx} f\left(x_2, y_\lambda^*(x_2)\right)\| \\
\leq & \|\nabla^2_{xx} f\left(x_1, y_\lambda^*(x_1)\right) - \nabla^2_{xx} f\left(x_2, y_\lambda^*(x_2)\right)\| \\
& + \|\nabla y_\lambda^*(x_1) \nabla^2_{yx} f\left(x_1, y_\lambda^*(x_1)\right) - \nabla y_\lambda^*(x_2) \nabla^2_{yx} f\left(x_1, y_\lambda^*(x_1)\right)\| \\
& + \|\nabla y_\lambda^*(x_2) \nabla^2_{yx} f\left(x_1, y_\lambda^*(x_1)\right) - \nabla y_\lambda^*(x_2) \nabla^2_{yx} f\left(x_2, y_\lambda^*(x_2)\right)\| \\
\leq & H_f(1 + \frac{4L_g}{\mu})^{\nu_f} \|x_1 - x_2\|^{\nu_f} + \frac{4L_g}{\mu} \rho_f(1 + \frac{4L_g}{\mu})\|x_1 - x_2\| \\
& + (1 + \frac{4L_g}{\mu})^2 (\frac{2\rho_g}{\mu} + \frac{\rho_f}{L_f}) L_f \|x_1 - x_2\| \\
\leq & \underbrace{H_f(1 + \frac{4L_g}{\mu})^{\nu_f}}_{C_1} \|x_1 - x_2\|^{\nu_f} \\
& + \underbrace{\left(\frac{4L_g}{\mu} \rho_f(1 + \frac{4L_g}{\mu}) + (1 + \frac{4L_g}{\mu})^2 (\frac{2\rho_g}{\mu} + \frac{\rho_f}{L_f}) L_f\right)}_{C_2} \mathcal{D}^{1-\nu_f} \|x_1 - x_2\|^{\nu_f}.
\end{aligned} \tag{15}$$

The first step applies the triangle inequality. The second step relies on the $(\nu_f, H_f)$-Hölder continuity of $\nabla^2_{xx} f$, the bound $\nabla^2_{yx} f(\cdot, \cdot) \preceq L_f$, and Lemma C.2. Here, $C_1 = \mathcal{O}(\ell \kappa^{\nu_f})$, $C_2 = \mathcal{O}(\ell \kappa^3)$.

Next, we evaluate $\nabla B(x)$ by differentiating:

$$\begin{aligned}
\nabla B(x) = & \lambda \left(\nabla^3_{xxx} g\left(x, y_\lambda^*(x)\right) - \nabla^3_{xxx} g\left(x, y^*(x)\right)\right) \\
& + \lambda \left(\nabla^3_{yxx} g\left(x, y_\lambda^*(x)\right) \times_1 \nabla y_\lambda^*(x) - \nabla^3_{yxx} g\left(x, y^*(x)\right) \times_1 \nabla y^*(x)\right) \\
& + \lambda \left(\nabla^3_{xyx} g\left(x, y_\lambda^*(x)\right) \times_2 \nabla y_\lambda^*(x) - \nabla^3_{xyx} g\left(x, y^*(x)\right) \times_2 \nabla y^*(x)\right) \\
& + \lambda \left(\nabla^3_{yyx} g\left(x, y_\lambda^*(x)\right) \times_1 \nabla y_\lambda^*(x) \times_2 \nabla y_\lambda^*(x) - \nabla^3_{yyx} g\left(x, y^*(x)\right) \times_1 \nabla y^*(x) \times_2 \nabla y^*(x)\right) \\
& + \lambda \left(\nabla^2 y_\lambda^*(x) \times_3 \left[\nabla^2_{yx} g\left(x, y_\lambda^*(x)\right)\right]^\top - \nabla^2 y^*(x) \times_3 \left[\nabla^2_{yx} g\left(x, y^*(x)\right)\right]^\top\right).
\end{aligned}$$

To bound the Lipschitz constant of $B(x)$, we control $\|\nabla B(x)\|$ as follows:

$$
\begin{aligned}
\|\nabla B(x)\| \leq & \lambda\|\nabla^3_{xxx}g(x,y^*(x)) - \nabla^3_{xxx}g(x,y^*_\lambda(x))\| \\
& + \lambda\|\nabla y^*(x)\|\|\nabla^3_{yxx}g(x,y^*(x)) - \nabla^3_{yxx}g(x,y^*_\lambda(x))\| \\
& + \lambda\|\nabla y^*_\lambda(x) - \nabla y^*(x)\|\|\nabla^3_{yxx}g(x,y^*_\lambda(x))\| \\
& + \lambda\|\nabla y^*(x)\|\|\nabla^3_{xyx}g(x,y^*(x)) - \nabla^3_{xyx}g(x,y^*_\lambda(x))\| \\
& + \lambda\|\nabla y^*(x) - \nabla y^*_\lambda(x)\|\|\nabla^3_{xyx}g(x,y^*_\lambda(x))\| \\
& + \lambda\|\nabla y^*(x)\|\|\nabla^3_{yyx}g(x,y^*(x))\|\|\nabla y^*_\lambda(x) - \nabla y^*(x)\| \\
& + \lambda\|\nabla y^*_\lambda(x)\|\|\nabla^3_{yyx}g(x,y^*(x))\|\|\nabla y^*_\lambda(x) - \nabla y^*(x)\| \\
& + \lambda\|\nabla y^*(x)\|^2\|\nabla^3_{yyx}g(x,y^*(x)) - \nabla^3_{yyx}g(x,y^*_\lambda(x))\| \\
& + \lambda\|\nabla^2 y^*(x)\|\|\nabla^2_{yx}g(x,y^*(x)) - \nabla^2_{yx}g(x,y^*_\lambda(x))\| \\
& + \lambda\|\nabla^2 y^*(x) - \nabla^2 y^*_\lambda(x)\|\|\nabla^2_{yx}g(x,y^*_\lambda(x))\|.
\end{aligned}
$$

Using the smoothness and Hölder continuity assumptions on $g$, as well as bounds from Lemma C.1, Lemma C.2, and Lemma C.4, we arrive at:

$$
\begin{aligned}
\|\nabla B(x)\| \leq & \lambda M_g\left(\frac{C_f}{\lambda\mu}\right)^{\nu_g}\left(1 + \frac{L_g}{\mu}\right)^2 + (2 + \frac{5L_g}{\mu})\lambda\rho_g\frac{D_2}{\lambda} \\
& + \lambda\rho_g\left(\frac{C_f}{\lambda\mu}\right)\left(1 + \frac{L_g}{\mu}\right)^2\frac{\rho_g}{\mu} + \lambda L_g\frac{D_4}{\lambda^{\nu_g}} \\
= & \lambda^{1-\nu_g} M_g\left(\frac{C_f}{\mu}\right)^{\nu_g}\left(1 + \frac{L_g}{\mu}\right)^2 + (2 + \frac{5L_g}{\mu})\rho_g D_2 \\
& + \rho_g\left(\frac{C_f}{\mu}\right)\left(1 + \frac{L_g}{\mu}\right)^2\frac{\rho_g}{\mu} + \lambda^{1-\nu_g} L_g D_4.
\end{aligned}
$$

Denote the entire right-hand side as $C_3 = \mathcal{O}(\lambda^{1-\nu_g}\ell\kappa^{4+\nu_g})$. Finally, we estimate the restricted Hölder constant of $\nabla^2 L^*_\lambda(x)$:

$$
\begin{aligned}
\frac{\|\nabla^2 L^*_\lambda(x_1) - \nabla^2 L^*_\lambda(x_2)\|}{\|x_1 - x_2\|^{\nu_f}} \leq & \frac{\|A(x_1) - A(x_2)\|}{\|x_1 - x_2\|^{\nu_f}} + \frac{\|B(x_1) - B(x_2)\|}{\|x_1 - x_2\|^{\nu_f}} \\
\leq & C_1 + (C_2 + C_3)\|x_1 - x_2\|^{1-\nu_f} \\
\leq & C_1 + (C_2 + C_3)\mathcal{R}^{1-\nu_f}.
\end{aligned}
$$

Define

$$
H_\nu(\lambda, \mathcal{R}) := C_1 + (C_2 + C_3)\mathcal{R}^{1-\nu_f} = \mathcal{O}(\ell\kappa^{\nu_f}) + \mathcal{O}(\lambda^{1-\nu_g}\ell\kappa^{4+\nu_g})\mathcal{R}^{1-\nu_f}. \tag{16}
$$

Thus, $\nabla^2 L^*_\lambda(x)$ is restrictively $(\nu_f, H_\nu(\lambda, \mathcal{R}))$-Hölder continuous with diameter $\mathcal{R}$. In the case $\nu_f = 1$ and $\nu_g = 1$, this implies $\nabla^2 L^\star_\lambda(x)$ is $\mathcal{O}(\ell\kappa^5)$-Lipschitz continuous. $\qquad\square$

# D  PROOF OF LEMMAS IN SECTION 3

## D.1  AGD SUBROUTINES

This method boasts an optimal convergence rate as shown below:

**Lemma D.1** ([16], Section 2). *Running Algorithm 2 on an $\ell_h$-smooth and $\mu_h$-strongly convex objective function $h(\cdot)$ with $\alpha = 1/\ell_h$ and $\beta = \left(\sqrt{\kappa_h} - 1\right) / \left(\sqrt{\kappa_h} + 1\right)$ produces an output $z_T$ satisfying*

$$
\|z_T - z^*\|^2 \leq (1 + \kappa_h)\left(1 - \frac{1}{\sqrt{\kappa_h}}\right)^T\|z_0 - z^*\|^2,
$$

*where $z^* = \arg\min_z h(z)$ and $\kappa_h = \ell_h/\mu_h$ denotes the condition number of the objective $h$.*

---

**Algorithm 2** AGD $(h, z_0, T, \alpha, \beta)$

---

1: **Input:** objective function $h(\cdot)$; start point $z_0$; iteration number $T \geq 1$; step-size $\alpha > 0$; momentum parameter $\beta \in (0, 1)$
2: $\tilde{z}_0 \leftarrow z_0$
3: **for** $t = 0, \ldots, T - 1$ **do**
4: $\quad z_{t+1} \leftarrow \tilde{z}_t - \alpha \nabla h(\tilde{z}_t)$
5: $\quad \tilde{z}_{t+1} \leftarrow z_{t+1} + \beta(z_{t+1} - z_t)$
6: **end for**
7: **Output:** $z_T$

---

### D.2 PROOF OF LEMMA 3

*Proof.* Consider an epoch ending at iteration $k \geq 2$. By applying the Cauchy–Schwarz inequality to the restart condition equation 5, we obtain

$$\max_{0 \leq i \leq j \leq k-1} \|x_i - x_j\| \leq \sum_{i=1}^{k-1} \|x_i - x_{i-1}\| \leq \sqrt{k S_{k-1}} \leq \left(\frac{L}{H_\nu}\right)^{\frac{1}{\nu_f}}. \tag{17}$$

This implies that the diameter of $\text{conv}(\{x_i\}_{i=0}^{k-1})$ is less than $\left(\frac{L}{H_\nu}\right)^{\frac{1}{\nu_f}}$. By solving a system of equations:

$$\begin{cases} \mathcal{R} = 3\left(\frac{L}{H_\nu}\right)^{\frac{1}{\nu_f}}, \\ H_\nu(\lambda, \mathcal{R}) = H_\nu, \end{cases} \tag{18}$$

where $H_\nu(\lambda, \mathcal{R})$ is defined in equation 16. We have

$$H_\nu = \mathcal{O}\left(\lambda^{\nu_f(1-\nu_g)} \ell \kappa^{3+(1+\nu_g)\nu_f}\right), \quad \mathcal{R} = \mathcal{O}\left(\lambda^{-(1-\nu_g)} \kappa^{-(1+\nu_g)}\right). \tag{19}$$

Denote this specific $\mathcal{R}$ by $\mathcal{D}$. The boundedness of $\{x_i\}_{i=1}^{k-1}$ has been ensured by equation 17. From line 8 in Algorithm 1, we have

$$\|w_{i+1} - w_i\| \leq (1 + \theta_{i+1})\|x_{i+1} - x_i\| + \theta_i \|x_i - x_{i-1}\| \leq 2\|x_{i+1} - x_i\| + \|x_i - x_{i-1}\|.$$

The last inequality holds due to $\theta_k \in (0, 1)$. So

$$\max_{0 \leq i < k} \|w_i - \bar{w}_k\| \leq \max_{0 \leq i \leq j < k} \|w_i - w_j\| \leq 3 \max_{0 \leq i \leq j < k} \|x_i - x_j\| \leq \mathcal{D},$$

where $\bar{w}_k$ is defined in equation 7. The first inequality holds because $\bar{w}_k \in \text{conv}(\{w_i\}_{i=0}^{k-1})$, and the maximum diameter of the convex hull is attained by a pair of its vertices.

$\square$

### D.3 PROOF OF LEMMA 4

*Proof.* Consider the exact gradient of $L_\lambda^*(\cdot)$:

$$\nabla L_\lambda^*(w_{t,k}) = \nabla_x f(w_{t,k}, y_\lambda^*(w_{t,k})) + \lambda \left(\nabla_x g(w_{t,k}, y_\lambda^*(w_{t,k})) - \nabla_x g(w_{t,k}, y^*(w_{t,k}))\right),$$

and the inexact gradient estimator used by Algorithm 1:

$$\hat{\nabla} L_\lambda^*(w_{t,k}) = \nabla_x f(w_{t,k}, y_{t,k}) + \lambda \left(\nabla_x g(w_{t,k}, y_{t,k}) - \nabla_x g(w_{t,k}, z_{t,k})\right).$$

By the triangle inequality, the Lipschitz continuity assumptions in Condition 1, and the condition $L_f \leq \frac{1}{2}\lambda\mu \leq \lambda L_g$, we obtain:

$$\begin{aligned} &\|\nabla L_\lambda^*(w_{t,k}) - \hat{\nabla} L_\lambda^*(w_{t,k})\| \\ \leq &L_f \|y_{t,k} - y_\lambda^*(w_{t,k})\| + \lambda L_g \|y_{t,k} - y_\lambda^*(w_{t,k})\| + \lambda L_g \|z_{t,k} - y^*(w_{t,k})\| \\ = &(L_f + \lambda L_g)\|y_{t,k} - y_\lambda^*(w_{t,k})\| + \lambda L_g \|z_{t,k} - y^*(w_{t,k})\| \\ \leq &(L_f + \lambda L_g) \cdot \frac{\sigma}{4\lambda L_g} + \lambda L_g \cdot \frac{\sigma}{2\lambda L_g} \\ \leq &\frac{\sigma}{2} + \frac{\sigma}{2} = \sigma. \end{aligned}$$

$\square$

# E    PROOF OF LEMMAS IN SECTION 4

**Lemma E.1.** *Under Assumption 1 and with $\lambda \geq 2L_f/\mu$, the following holds for any $x$ and $x'$:*

$$L_\lambda^*(x) - L_\lambda^*(x') \leq \langle \nabla L_\lambda^*(x'), x - x' \rangle + \frac{L}{2}\|x - x'\|^2.$$

## E.1    PROOF OF LEMMA 5

*Proof.* Let $\bar{x} = \sum_{i=1}^n q_i x_i$. Since $L_\lambda^*$ is twice differentiable, we have

$$\nabla L_\lambda^*(x_i) - \nabla L_\lambda^*(\bar{x}) = \nabla^2 L_\lambda(\bar{x})(x_i - \bar{x}) + \int_0^1 (\nabla^2 L_\lambda^*(\bar{x} + t(x_i - \bar{x})) - \nabla^2 L_\lambda^*(\bar{x}))(x_i - \bar{x})\,\mathrm{d}t.$$

Computing the weighted average sum, we have

$$\sum_{i=1}^n q_i \nabla L_\lambda^*(x_i) - \nabla L_\lambda^*(\bar{x}) = \sum_{i=1}^n q_i \int_0^1 (\nabla^2 L_\lambda^*(\bar{x} + t(x_i - \bar{x})) - \nabla^2 L_\lambda^*(\bar{x}))(x_i - \bar{x})\,\mathrm{d}t$$

and

$$\left\| \sum_{i=1}^n q_i \nabla L_\lambda^*(x_i) - \nabla L_\lambda^*(\bar{x}) \right\| \leq \sum_{i=1}^n q_i \int_0^1 \left\| \nabla^2 L_\lambda^*(\bar{x} + t(x_i - \bar{x})) - \nabla^2 L_\lambda^*(\bar{x}) \right\| \|x_i - \bar{x}\|\,\mathrm{d}t$$

$$\leq \sum_{i=1}^n q_i \int_0^1 H_\nu \|t(x_i - \bar{x})\|^{\nu_f} \|x_i - \bar{x}\|\,\mathrm{d}t$$

$$= \frac{H_\nu}{1 + \nu_f} \sum_{i=1}^n q_i \|x_i - \bar{x}\|^{1+\nu_f}$$

$$= \frac{H_\nu}{1 + \nu_f} \sum_{i=1}^n q_i^{\frac{1-\nu_f}{2}} \left( q_i \|x_i - \bar{x}\|^2 \right)^{\frac{1+\nu_f}{2}}$$

$$\leq \frac{H_\nu}{1 + \nu_f} \left( \sum_{i=1}^n q_i \right)^{\frac{1-\nu_f}{2}} \left( \sum_{i=1}^n q_i \|x_i - \bar{x}\|^2 \right)^{\frac{1+\nu_f}{2}}$$

$$= \frac{H_\nu}{1 + \nu_f} \left( \sum_{1 \leq i < j \leq n} q_i q_j \|x_i - x_j\|^2 \right)^{\frac{1+\nu_f}{2}}.$$

The second inequality holds due to $\|x_i - \bar{x}\| \leq \max_{1 \leq i \leq j \leq n} \|x_i - x_j\| \leq \mathcal{D}$, Lemma 2 and equation equation 6. The last inequality uses Hölder inequality. The last equality holds due to $\sum_{i=1}^n q_i = 1$ and $\sum_{i=1}^n q_i \|x_i - \bar{x}\|^2 = \sum_{1 \leq i < j \leq n} q_i q_j \|x_i - x_j\|^2$. $\qquad \square$

## E.2    PROOF OF LEMMA 6

*Proof.*

$$L_\lambda^*(x) - L_\lambda^*(x') - \frac{1}{2}\langle \nabla L_\lambda^*(x) + \nabla L_\lambda^*(x'), x - x' \rangle$$

$$= \int_0^1 \langle \nabla L_\lambda^*(tx + (1-t)x'), x - x' \rangle - \frac{1}{2}\langle \nabla L_\lambda^*(x) + \nabla L_\lambda^*(x'), x - x' \rangle\,\mathrm{d}t$$

$$= \int_0^1 \langle \nabla L_\lambda^*(tx + (1-t)x') - t\nabla L_\lambda^*(x) - (1-t)\nabla L_\lambda^*(x'), x - x' \rangle\,\mathrm{d}t$$

$$\leq \int_0^1 \|\nabla L_\lambda^*(tx + (1-t)x') - t\nabla L_\lambda^*(x) - (1-t)\nabla L_\lambda^*(x')\| \|x - x'\|\,\mathrm{d}t$$

$$\leq \frac{H_\nu}{1 + \nu_f} \int_0^1 \left( t(1-t)^{1+\nu_f} + (1-t)t^{1+\nu_f} \right) \|x - x'\|^{2+\nu_f}\,\mathrm{d}t$$

$$= \frac{2H_\nu}{(1 + \nu_f)(2 + \nu_f)(3 + \nu_f)} \|x - x'\|^{2+\nu_f}.$$

The last inequality follows from Lemma 5 by setting $n = 2$, $(x_1, x_2) = (x, x')$, and $(q_1, q_2) = (t, 1-t)$.

$\square$

### E.3 PROOF OF LEMMA 7

*Proof.* Let

$$P_k := \langle \nabla L_\lambda^*(x_{k-1}), x_k - x_{k-1} \rangle.$$

From Lemma E.1, we have

$$L_\lambda^*(x_{k+1}) - L_\lambda^*(w_k) \leq \langle \nabla L_\lambda^*(w_k), x_{k+1} - w_k \rangle + \frac{L}{2} \|x_{k+1} - w_k\|^2$$
$$= -\frac{1}{L} \langle \nabla L_\lambda^*(w_k), \hat{\nabla} L_\lambda^*(w_k) \rangle + \frac{1}{2L} \|\hat{\nabla} L_\lambda^*(w_k)\|^2. \tag{20}$$

From Lemma 6 and Lemma 3, it follows that $\|w_k - x_k\| \leq \|x_k - x_{k-1}\| \leq \mathcal{D}$ and

$$L_\lambda^*(w_k) - L_\lambda^*(x_k) \leq \frac{1}{2} \langle \nabla L_\lambda^*(w_k) + \nabla L_\lambda^*(x_k), w_k - x_k \rangle$$
$$+ \frac{2H_\nu}{(1+\nu_f)(2+\nu_f)(3+\nu_f)} \|w_k - x_k\|^{2+\nu_f}. \tag{21}$$

By summing inequalities equation 20 and equation 21, we evaluate the expression as follows

$$L_\lambda^*(x_{k+1}) - L_\lambda^*(x_k)$$
$$\leq \frac{1}{2} \langle \nabla L_\lambda^*(w_k) + \nabla L_\lambda^*(x_k), w_k - x_k \rangle + \frac{2H_\nu \theta_k^{2+\nu_f}}{(1+\nu_f)(2+\nu_f)(3+\nu_f)} \|x_k - x_{k-1}\|^{2+\nu_f}$$
$$- \frac{1}{L} \langle \nabla L_\lambda^*(w_k), \hat{\nabla} L_\lambda^*(w_k) \rangle + \frac{1}{2L} \|\hat{\nabla} L_\lambda^*(w_k)\|^2. \tag{22}$$

To evaluate the first term on the right-hand side, we decompose it into four terms:

$$\langle \nabla L_\lambda^*(w_k) + \nabla L_\lambda^*(x_k), w_k - x_k \rangle$$
$$= \underbrace{2\langle \nabla L_\lambda^*(w_k), w_k - x_k \rangle}_{(A)} + \underbrace{\theta_k \langle \nabla L_\lambda^*(x_{k-1}), w_k - x_k \rangle}_{(B)}$$
$$\underbrace{-\theta_k \langle \nabla L_\lambda^*(x_k), w_k - x_k \rangle}_{(C)} \underbrace{-\langle \nabla L_\lambda^*(w_k) + \theta_k \nabla L_\lambda^*(x_{k-1}) - (1+\theta_k) \nabla L_\lambda^*(x_k), w_k - x_k \rangle}_{(D)}.$$

Let $n = 2$, $q_1 = 1/(1+\theta_k)$, $q_2 = \theta_k/(1+\theta_k)$ in Lemma 5, we have

$$\left\| \nabla L_\lambda^*(x_k) - \frac{1}{1+\theta_k} \nabla L_\lambda^*(w_k) - \frac{\theta_k}{1+\theta_k} \nabla L_\lambda^*(x_{k-1}) \right\|$$
$$\leq \frac{H_\nu}{1+\nu_f} \left( \frac{\theta_k}{(1+\theta_k)^2} \|w_k - x_{k-1}\|^2 \right)^{\frac{1+\nu_f}{2}}$$
$$= \frac{H_\nu}{1+\nu_f} \theta_k^{\frac{1+\nu_f}{2}} \|x_k - x_{k-1}\|^{1+\nu_f}. \tag{23}$$

Now, we proceed to evaluate (A), (B), (C) and (D) respectively.

$$(A) = \frac{1}{L}\|\nabla L_\lambda^*(w_k)\|^2 + L\|w_k - x_k\|^2 - L\|(w_k - x_k) - \frac{1}{L}\nabla L_\lambda^*(w_k)\|^2$$

$$= \frac{1}{L}\|\nabla L_\lambda^*(w_k)\|^2 + \theta_k^2 L\|x_k - x_{k-1}\|^2 - L\left\|(x_{k+1} - x_k) + \left(\frac{1}{L}\hat\nabla L_\lambda^*(w_k) - \frac{1}{L}\nabla L_\lambda^*(w_k)\right)\right\|^2$$

$$= \frac{1}{L}\|\nabla L_\lambda^*(w_k)\|^2 + \theta_k^2 L\|x_k - x_{k-1}\|^2 - L\|x_{k+1} - x_k\|^2$$

$$- \frac{1}{L}\left\|\hat\nabla L_\lambda^*(w_k) - \nabla L_\lambda^*(w_k)\right\|^2 - 2\langle x_{k+1} - x_k, \hat\nabla L_\lambda^*(w_k) - \nabla L_\lambda^*(w_k)\rangle,$$

$$(B) = \theta_k^2\langle\nabla L_\lambda^*(x_{k-1}), x_k - x_{k-1}\rangle = \theta_k^2 P_k,$$

$$(C) = -\theta_k P_{k+1} + \theta_k\langle\nabla L_\lambda^*(x_k), x_{k+1} - w_k\rangle$$

$$= -\theta_k P_{k+1} - \frac{\theta_k}{L}\langle\nabla L_\lambda^*(x_k), \hat\nabla L_\lambda^*(w_k)\rangle,$$

$$(D) \le \frac{2H_\nu}{1+\nu_f}\theta_k^{\frac{3+\nu_f}{2}}\|x_k - x_{k-1}\|^{2+\nu_f}.$$

Here we use equality $2\langle a, b\rangle = \frac{1}{L}\|a\|^2 + L\|b\|^2 - L\|b - \frac{1}{L}a\|^2$, $x_{k+1} = w_k - \frac{1}{L}\hat\nabla L_\lambda^*(w_k)$, $w_k = x_k + \theta_k(x_k - x_{k-1})$ and equation 23. Plugging the evaluations into (22), we have

$$L_\lambda^*(x_{k+1}) - L_\lambda^*(x_k) \le \frac{2H_\nu}{(1+\nu_f)(2+\nu_f)(3+\nu_f)}\theta_k^{2+\nu_f}\|x_k - x_{k-1}\|^{2+\nu_f}$$

$$+ \frac{\theta_k^2 L}{2}\|x_k - x_{k-1}\|^2 - \frac{L}{2}\|x_{k+1} - x_k\|^2$$

$$- \langle x_{k+1} - x_k, \hat\nabla L_\lambda^*(w_k) - \nabla L_\lambda^*(w_k)\rangle$$

$$+ \frac{\theta_k^2}{2}P_k - \frac{\theta_k}{2}P_{k+1} + \frac{H_\nu}{1+\nu_f}\theta_k^{\frac{3+\nu_f}{2}}\|x_k - x_{k-1}\|^{2+\nu_f}$$

$$- \frac{\theta_k}{2L}\langle\nabla L_\lambda^*(x_k), \hat\nabla L_\lambda^*(w_k)\rangle. \tag{24}$$

Next, to bound the last term on the right-hand side of equation 24, by triangle inequality and equation 23, we have

$$\left\|(1+\theta_k)\nabla L_\lambda^*(x_k) - \hat\nabla L_\lambda^*(w_k)\right\|$$

$$\le \|(1+\theta_k)\nabla L_\lambda^*(x_k) - \nabla L_\lambda^*(w_k)\| + \left\|\hat\nabla L_\lambda^*(w_k) - \nabla L_\lambda^*(w_k)\right\|$$

$$\le \sigma + \theta_k\|\nabla L_\lambda^*(x_{k-1})\| + \frac{2H_\nu}{1+\nu_f}\theta_k^{\frac{1+\nu_f}{2}}\|x_k - x_{k-1}\|^{1+\nu_f}.$$

Squaring both sides yields

$$\|(1+\theta_k)\nabla L_\lambda^*(x_k) - \hat\nabla L_\lambda^*(w_k)\|^2$$

$$= (1+\theta_k)^2\|\nabla L_\lambda^*(x_k)\|^2 + \|\hat\nabla L_\lambda^*(w_k)\|^2 - 2(1+\theta_k)\langle\nabla L_\lambda^*(x_k), \hat\nabla L_\lambda^*(w_k)\rangle$$

$$\ge (1+\theta_k)^2\|\nabla L_\lambda^*(x_k)\|^2 - 2(1+\theta_k)\langle\nabla L_\lambda^*(x_k), \hat\nabla L_\lambda^*(w_k)\rangle,$$

and

$$\left(\sigma + \theta_k\|\nabla L_\lambda^*(x_{k-1})\| + \frac{2H_\nu}{1+\nu_f}\theta_k^{\frac{1+\nu_f}{2}}\|x_k - x_{k-1}\|^{1+\nu_f}\right)^2$$

$$\le \theta_k(1+\theta_k)\|\nabla L_\lambda^*(x_{k-1})\|^2 + 2(1+\theta_k)\left(\sigma^2 + \frac{4H_\nu^2}{(1+\nu_f)^2}\theta_k^{1+\nu_f}\|x_k - x_{k-1}\|^{2+2\nu_f}\right).$$

Here we use the inequalities $(a+b)^2 \le (1+\frac{1}{\theta_k})a^2 + (1+\theta_k)b^2$ and $(a+b)^2 \le 2(a^2+b^2)$. Rearranging the terms yields

$$-\langle\nabla L_\lambda^*(x_k), \hat\nabla L_\lambda^*(w_k)\rangle \le \sigma^2 + \frac{\theta_k}{2}\|\nabla L_\lambda^*(x_{k-1})\|^2 + \frac{4H_\nu^2}{(1+\nu_f)^2}\theta_k^{1+\nu_f}\|x_k - x_{k-1}\|^{2+2\nu_f}$$

$$- \frac{1+\theta_k}{2}\|\nabla L_\lambda^*(x_k)\|^2.$$

By plugging this bound into (24): we obtain

$$
\begin{aligned}
L_\lambda^*(x_{k+1}) - L_\lambda^*(x_k) \leq & \frac{2H_\nu}{(1+\nu_f)(2+\nu_f)(3+\nu_f)}\theta_k^{2+\nu_f}\|x_k - x_{k-1}\|^{2+\nu_f} \\
& + \frac{\theta_k^2 L}{2}\|x_k - x_{k-1}\|^2 - \frac{L}{2}\|x_{k+1} - x_k\|^2 \\
& - \langle x_{k+1} - x_k, \hat{\nabla}L_\lambda^*(w_k) - \nabla L_\lambda^*(w_k)\rangle \\
& + \frac{\theta_k^2}{2}P_k - \frac{\theta_k}{2}P_{k+1} + \frac{H_\nu}{1+\nu_f}\theta_k^{\frac{3+\nu_f}{2}}\|x_k - x_{k-1}\|^{2+\nu_f} \\
& + \frac{\theta_k^2}{4L}\|\nabla L_\lambda^*(x_{k-1})\|^2 + \frac{2H_\nu^2}{(1+\nu_f)^2}\frac{\theta_k^{2+\nu_f}}{L}\|x_k - x_{k-1}\|^{2+2\nu_f} \\
& - \frac{(1+\theta_k)\theta_k}{4L}\|\nabla L_\lambda^*(x_k)\|^2 + \frac{\theta_k \sigma^2}{2L}.
\end{aligned}
\tag{25}
$$

Considering equation 9, equation 25 and $\theta_k \leq 1$, we have

$$
\begin{aligned}
\Phi_{k+1} - \Phi_k \leq & L_\lambda^*(x_{k+1}) - L_\lambda^*(x_k) + \frac{\theta_{k+1}^2}{2}\left(P_{k+1} + \frac{1}{2L}\|\nabla L_\lambda^*(x_k)\|^2 + L\|x_{k+1} - x_k\|^2\right) \\
& - \frac{\theta_k^2}{2}\left(P_k + \frac{1}{2L}\|\nabla L_\lambda^*(x_{k-1})\|^2 + L\|x_k - x_{k-1}\|^2\right) \\
\leq & \|x_k - x_{k-1}\|^{2+\nu_f}\left(\frac{2H_\nu}{(1+\nu_f)(2+\nu_f)(3+\nu_f)}\theta_k^{2+\nu_f} + \frac{H_\nu}{1+\nu_f}\theta_k^{\frac{3+\nu_f}{2}}\right) \\
& + \|x_k - x_{k-1}\|^{2+2\nu_f}\frac{2H_\nu^2}{(1+\nu_f)^2}\frac{\theta_k^{2+\nu_f}}{L} + \frac{\theta_{k+1}^2 - \theta_k}{2}P_{k+1} \\
& + \frac{\theta_{k+1}^2 - \theta_k(1+\theta_k)}{4L}\|\nabla L_\lambda^*(x_k)\|^2 + \frac{\sigma^2}{2L} + \sigma\|x_{k+1} - x_k\|.
\end{aligned}
$$

From Young's inequalities and $\theta_{k+1}^2 - \theta_k \leq 0$, we have

$$
-P_{k+1} = -\langle \nabla L_\lambda^*(x_k), x_{k+1} - x_k\rangle \leq \frac{1}{2L}\|\nabla L_\lambda^*(x_k)\|^2 + \frac{L}{2}\|x_{k+1} - x_k\|^2.
$$

Finally, we derive the inequality below:

$$
\begin{aligned}
\Phi_{k+1} - \Phi_k \leq & \|x_k - x_{k-1}\|^{2+\nu_f}\left(\frac{2H_\nu}{(1+\nu_f)(2+\nu_f)(3+\nu_f)}\theta_k^{2+\nu_f} + \frac{H_\nu}{1+\nu_f}\theta_k^{\frac{3+\nu_f}{2}}\right) \\
& + \|x_k - x_{k-1}\|^{2+2\nu_f}\frac{2H_\nu^2}{(1+\nu_f)^2}\frac{\theta_k^{2+\nu_f}}{L} + \frac{\theta_{k+1}^2 + \theta_k - 2}{4}L\|x_{k+1} - x_k\|^2 \\
& - \frac{\theta_k^2}{4L}\|\nabla L_\lambda^*(x_k)\|^2 + \frac{\sigma^2}{2L} + \sigma\|x_{k+1} - x_k\|.
\end{aligned}
$$

$\square$

### E.4 PROOF OF LEMMA 8

*Proof.* Summing Lemma 7 from $i = 0, \ldots, k-1$ and telescoping yields

$$
\Phi_k - \Phi_0 = \sum_{i=0}^{k-1} (\Phi_{i+1} - \Phi_i)
$$

$$
\leq \sum_{i=0}^{k-1} \left( \|x_i - x_{i-1}\|^{2+\nu_f} \left( \frac{2H_\nu}{(1+\nu_f)(2+\nu_f)(3+\nu_f)} \theta_i^{2+\nu_f} + \frac{H_\nu}{1+\nu_f} \theta_i^{\frac{3+\nu_f}{2}} \right) \right.
$$

$$
+ \|x_i - x_{i-1}\|^{2+2\nu_f} \frac{2H_\nu^2}{(1+\nu_f)^2} \frac{\theta_i^{2+\nu_f}}{L} + \frac{\theta_{i+1}^2 + \theta_i - 2}{4} L\|x_{i+1} - x_i\|^2
$$

$$
\left. - \frac{\theta_i^2}{4L} \|\nabla L_\lambda^*(x_i)\|^2 + \frac{\sigma^2}{2L} + \sigma\|x_{i+1} - x_i\| \right)
$$

$$
\leq \sum_{i=0}^{k-1} \|x_i - x_{i-1}\|^{2+\nu_f} \left( \frac{2H_\nu}{(1+\nu_f)(2+\nu_f)(3+\nu_f)} \theta_{k-1}^{2+\nu_f} + \frac{H_\nu}{1+\nu_f} \theta_{k-1}^{\frac{3+\nu_f}{2}} \right)
$$

$$
+ \sum_{i=0}^{k-1} \|x_i - x_{i-1}\|^{2+2\nu_f} \frac{2H_\nu^2}{(1+\nu_f)^2} \frac{\theta_{k-1}^{2+\nu_f}}{L} + \frac{\theta_k^2 + \theta_{k-1} - 2}{4} L \sum_{i=0}^{k-1} \|x_{i+1} - x_i\|^2
$$

$$
- \frac{\theta_0^2}{4L} \|\nabla L_\lambda^*(x_i)\|^2 + \frac{k\sigma^2}{2L} + \sigma \sum_{i=0}^{k-1} \|x_{i+1} - x_i\| \right). \tag{26}
$$

The second inequality holds due to $\{\theta_k\}$ is non-decreasing and non-negative. Moreover, by the definition of $\Phi_k$ in equation 9 , we have

$$
\Phi_k - L_\lambda^*(x_k) = \frac{\theta_k^2}{2} \left( \frac{1}{2L} \|\nabla L_\lambda^*(x_{k-1}) + L(x_k - x_{k-1})\|^2 + \frac{L}{2} \|x_k - x_{k-1}\|^2 \right) \geq 0, \tag{27}
$$

$$
\Phi_0 - L_\lambda^*(x_0) = \frac{\theta_0^2}{4L} \|L_\lambda^*(x_0)\|^2 \geq 0. \tag{28}
$$

From Power-Mean Inequality, we have

$$
\sum_{i=0}^{k-1} \|x_i - x_{i-1}\|^{2+\nu_f} \leq S_{k-1}^{\frac{2+\nu_f}{2}}, \quad \sum_{i=0}^{k-1} \|x_i - x_{i-1}\|^{2+2\nu_f} \leq S_{k-1}^{1+\nu_f}. \tag{29}
$$

Substituting equation 27, equation 28, and equation 29 into equation 26, we obtain

$$
L_\lambda^*(x_k) - L_\lambda^*(x_0) \leq S_{k-1}^{\frac{2+\nu_f}{2}} \left( \frac{2H_\nu}{(1+\nu_f)(2+\nu_f)(3+\nu_f)} \theta_{k-1}^{2+\nu_f} + \frac{H_\nu}{1+\nu_f} \theta_{k-1}^{\frac{3+\nu_f}{2}} \right)
$$

$$
+ S_{k-1}^{1+\nu_f} \cdot \frac{2H_\nu^2}{(1+\nu_f)^2} \cdot \frac{\theta_{k-1}^{2+\nu_f}}{L} + \frac{\theta_k^2 + \theta_{k-1} - 2}{4} LS_k
$$

$$
+ \frac{k\sigma^2}{2L} + \sigma \sum_{i=0}^{k-1} \|x_{i+1} - x_i\|.
$$

Applying the restart condition equation 5 and noting that $S_{k-1} \leq S_k$, we further obtain

$$
L_\lambda^*(x_k) - L_\lambda^*(x_0) \leq \left( \frac{2}{(1+\nu_f)(2+\nu_f)(3+\nu_f)} \theta_{k-1}^{2+\nu_f} + \frac{1}{1+\nu_f} \theta_{k-1}^{\frac{3+\nu_f}{2}} \right) \cdot \frac{LS_k}{k^{2+\frac{\nu_f}{2}}}
$$

$$
+ \frac{2}{(1+\nu_f)^2} \theta_{k-1}^{2+\nu_f} \cdot \frac{LS_k}{k^{4+\nu_f}} + \frac{\theta_k^2 + \theta_{k-1} - 2}{4} LS_k
$$

$$
+ \frac{k\sigma^2}{2L} + \sigma \sum_{i=0}^{k-1} \|x_{i+1} - x_i\|.
$$

Since $0 \le \nu_f \le 1$, and

$$\left(\frac{7}{3}\theta_{k-1}^{2+\nu_f} + \theta_{k-1}^{\frac{3+\nu_f}{2}}\right) \cdot \frac{1}{k^{2+\frac{\nu_f}{2}}} + \frac{\theta_k^2 + \theta_{k-1} - 2}{4} \le -\frac{1}{32k}, \quad \forall k \ge 1,$$

we obtain

$$L_\lambda^*(x_k) - L_\lambda^*(x_0) \le LS_k \left(\left(\frac{7}{3}\theta_{k-1}^{2+\nu_f} + \theta_{k-1}^{\frac{3+\nu_f}{2}}\right) \cdot \frac{1}{k^{2+\frac{\nu_f}{2}}} + \frac{\theta_k^2 + \theta_{k-1} - 2}{4}\right)$$

$$+ \frac{k\sigma^2}{2L} + \sigma \sum_{i=0}^{k-1} \|x_{i+1} - x_i\|$$

$$\le -\frac{LS_k}{32k} + \frac{k\sigma^2}{2L} + \sigma \sum_{i=0}^{k-1} \|x_{i+1} - x_i\|.$$

$\square$

### E.5 PROOF OF LEMMA 9

*Proof.* Define

$$Z_k = \sum_{i=0}^{k-1} \prod_{j=i+1}^{k-1} \theta_j = \frac{k+1}{2},$$

so that $p_{k,i} = \frac{1}{Z_k} \prod_{j=i+1}^{k-1} \theta_j$. From definition (7), we have:

$$\sum_{i=0}^{k-1} p_{k,i} \hat{\nabla} L_\lambda^*(w_i) = \sum_{i=0}^{k-1} p_{k,i} L(w_i - x_{i+1})$$

$$= \sum_{i=0}^{k-1} p_{k,i} L(\theta_i(x_i - x_{i-1}) - (x_{i+1} - x_i))$$

$$= \sum_{i=0}^{k-1} L\left(p_{k,i-1}(x_i - x_{i-1}) - p_{k,i}(x_{i+1} - x_i)\right)$$

$$= -Lp_{k,k-1}(x_k - x_{k-1}).$$

From $\bar{w}_k \in \text{conv}(\{w_i\}_{i=0}^{k-1})$, Lemma 3 and Lemma 5, we have

$$\|\nabla L_\lambda^*(\bar{w}_k)\| \le \left\|\sum_{i=0}^{k-1} p_{k,i} \nabla L_\lambda^*(w_i)\right\| + \frac{H_\nu}{1+\nu_f}\left(\sum_{0 \le i < j < k} p_{k,i} p_{k,j} \|w_i - w_j\|^2\right)^{\frac{1+\nu_f}{2}}$$

$$\le \sigma + Lp_{k,k-1}\|x_k - x_{k-1}\| + \frac{H_\nu}{1+\nu_f}\left(\sum_{0 \le i < j < k} p_{k,i} p_{k,j} \|w_i - w_j\|^2\right)^{\frac{1+\nu_f}{2}}$$

$$\le \sigma + \frac{L}{Z_k}\|x_k - x_{k-1}\| + \frac{H_\nu}{(1+\nu_f)Z_k^{1+\nu_f}}\left(\sum_{0 \le i < j < k} \|w_i - w_j\|^2\right)^{\frac{1+\nu_f}{2}}. \tag{30}$$

Here we use inequality $p_{k,i} \leq p_{k,k-1} = 1/Z_k = 2/(k+1)$ for all $0 \leq i < k$. Regarding the last term in equation 30, we have

$$\|w_i - w_j\|$$

$$\leq \|w_i - x_i\| + \sum_{l=i+1}^{j-1} \|x_l - x_{l-1}\| + \|w_j - x_{j-1}\|$$

$$= \|x_i - x_{i-1}\| + \sum_{l=i+1}^{j-1} \|x_l - x_{l-1}\| + 2\|x_j - x_{j-1}\|$$

$$\leq \left(1^2 + \sum_{l=i+1}^{j-1} 1^2 + 2^2\right)^{1/2} \left(\sum_{l=i}^{j} \|x_l - x_{l-1}\|^2\right)^{1/2}$$

$$= \sqrt{j - i + 4}\left(\sum_{l=i}^{j} \|x_l - x_{l-1}\|^2\right)^{1/2}.$$

The above inequalities hold by the triangle inequality, $0 \leq \theta_k \leq 1$ and Cauchy–Schwarz inequality, respectively. Then

$$\sum_{0 \leq i < j < k} \|w_i - w_j\|^2 \leq \sum_{0 \leq i < j < k} \sum_{l=i}^{j} (j - i + 4)\|x_l - x_{l-1}\|^2$$

$$= \sum_{l=0}^{k-1} \left(\sum_{i=0}^{l} \sum_{j=l}^{k-1} (j - i + 4)\right) \|x_l - x_{l-1}\|^2 - 4\sum_{l=0}^{k-1} \|x_l - x_{l-1}\|^2$$

$$= \frac{k+7}{2} \sum_{l=0}^{k-1} (l+1)(k-l)\|x_l - x_{l-1}\|^2 - 4\sum_{l=0}^{k-1} \|x_l - x_{l-1}\|^2$$

$$\leq \frac{k+7}{2} \sum_{l=0}^{k-1} \frac{(k+1)^2}{4}\|x_l - x_{l-1}\|^2 - 4\sum_{l=0}^{k-1} \|x_l - x_{l-1}\|^2$$

$$= \frac{(k-1)(k+5)^2}{8} \sum_{l=0}^{k-1} \|x_l - x_{l-1}\|^2 \leq \frac{(k-1)(k+5)^2}{8} S_k. \tag{31}$$

Plugging equation 31 into equation 30, we have

$$\|\nabla L_\lambda^*(\bar{w}_k)\| \leq \sigma + \frac{L}{Z_k}\|x_k - x_{k-1}\| + \frac{H_\nu}{1+\nu_f}(1/Z_k)^{1+\nu_f}\left(\frac{(k-1)(k+5)^2}{8}\right)^{\frac{1+\nu_f}{2}} S_k^{\frac{1+\nu_f}{2}}.$$

$$\tag{32}$$

Then for $k \geq 2$, combing with (32), we have

$$\left(\sum_{i=1}^{k-1} Z_i^2\right) \min_{1 \leq i < k} \|\nabla L_\lambda^*(\bar{w}_i)\|$$

$$\leq \sum_{i=1}^{k-1} Z_i^2 \|\nabla L_\lambda^*(\bar{w}_i)\|$$

$$\leq \sigma \sum_{i=1}^{k-1} Z_i^2 + \sum_{i=1}^{k-1} \left( LZ_i \|x_i - x_{i-1}\| + \frac{H_\nu}{1+\nu_f}(1/Z_i)^{\nu_f-1}(\frac{(i-1)(i+5)^2}{8})^{\frac{1+\nu_f}{2}} S_i^{\frac{1+\nu_f}{2}} \right)$$

$$\leq \sigma \sum_{i=1}^{k-1} Z_i^2 + L\sqrt{S_{k-1}}(\sum_{i=1}^{k-1} Z_i^2)^{1/2} + \frac{H_\nu}{1+\nu_f} \sum_{i=1}^{k-1}(1/Z_i)^{\nu_f-1}(\frac{(i-1)(i+5)^2}{8})^{\frac{1+\nu_f}{2}} S_{k-1}^{(1+\nu_f)/2}$$

$$\leq \sigma \sum_{i=1}^{k-1} Z_i^2 + L\sqrt{S_{k-1}}(\sum_{i=1}^{k-1} Z_i^2)^{1/2} + \frac{L\sqrt{1/k^{4+\nu_f}}}{1+\nu_f} \sum(\frac{2}{i+1})^{\nu_f-1}(\frac{(i-1)(i+5)^2}{8})^{\frac{1+\nu_f}{2}} S_{k-1}^{\frac{1}{2}}$$

$$= \sigma \sum_{i=1}^{k-1} Z_i^2 + L\sqrt{S_{k-1}} \left( (\sum_{i=1}^{k-1} Z_i^2)^{1/2} + \frac{\sqrt{1/k^{4+\nu_f}}}{1+\nu_f} \sum(\frac{2}{i+1})^{\nu_f-1}(\frac{(i-1)(i+5)^2}{8})^{\frac{1+\nu_f}{2}} \right).$$

Notice that $Z_k = \frac{k+1}{2}$ and $\frac{k^3}{12} \leq \sum Z_i^2 \leq \frac{k^3}{6}$, we have

$$\min_{1 \leq i < k} \|\nabla L_\lambda^*(\bar{w}_i)\| \leq \sigma + L\sqrt{S_{k-1}} \frac{\left( (\sum_{i=1}^{k-1} Z_i^2)^{1/2} + \frac{\sqrt{1/k^{4+\nu_f}}}{1+\nu_f} \sum(\frac{2}{i+1})^{\nu_f-1}(\frac{(i-1)(i+5)^2}{8})^{(1+\nu_f)/2} \right)}{\left( \sum_{i=1}^{k-1} Z_i^2 \right)}$$

$$\leq \sigma + L\sqrt{S_{k-1}} \frac{\frac{k^{\frac{3}{2}}}{\sqrt{6}} + \sqrt{\frac{1}{k^{4+\nu_f}}} \sum_{i=1}^{k-1} \frac{9}{2} i^{\frac{5}{2}+\frac{\nu_f}{2}}}{k^3/12}$$

$$\leq \sigma + cL\sqrt{S_{k-1}/k^3},$$

where c is a constant, $c = 2\sqrt{6} + 27$. The last inequality holds due to $\sum_{i=1}^{k-1} i^{\frac{5}{2}+\frac{\nu_f}{2}} \leq \frac{1}{2} k^{\frac{7}{2}+\frac{\nu_f}{2}}$. $\quad\square$

### E.6 PROOF OF PROPOSITION 1

*Proof.* Consider an epoch ends at iteration $k$ and ignore the subscript $t$. If $\bar{w}_k$ is not an $\epsilon$-first-order stationary point and $k \geq 2$, from Lemma 9, we have:

$$\epsilon \leq \sigma + cL\sqrt{S_{k-1}/k^3} \leq \sigma + cL\sqrt{S_k/k^3}.$$

If $k = 1$, $\sigma + cL\sqrt{S_k/k^3} = \sigma + cL\|x_1 - x_0\| = \sigma + c\|\hat{\nabla}L_\lambda^*(x_0)\| \geq \epsilon$. Here we set $\sigma = \frac{1}{64c+1}\epsilon$, the above inequality is

$$S_k \geq \frac{\epsilon^2 k^3}{\left(c + \frac{1}{64}\right)^2 L^2}, \qquad \forall\, k \geq 1. \tag{33}$$

From (33), We have

$$\sigma\sqrt{kS_k} = \frac{1}{64c+1}\epsilon\sqrt{kS_k} \leq \frac{LS_k}{64k}, \tag{34}$$

$$\frac{k\sigma^2}{2L} \leq \frac{k}{2L}\frac{1}{64^2}L^2\frac{S_k}{k^3} \leq \frac{LS_k}{2 \times 64^2 k}. \tag{35}$$

From restart condition equation 5, we have

$$S_k > \left(\frac{L^2/k^{4+\nu_f}}{H_\nu^2}\right)^{1/\nu_f}. \tag{36}$$

Then we can bound $S_k$ as:

$$S_k = S_k^{\frac{4+3\nu_f}{4+4\nu_f}} S_k^{\frac{\nu_f}{4+4\nu_f}} \geq L^{-\frac{3}{2}} \left(\frac{64\epsilon}{64c+1}\right)^{\frac{4+3\nu_f}{2+2\nu_f}} k^2 H_\nu^{-\frac{1}{2+2\nu_f}}.$$

From Lemma 8, (34) and (35), in this epoch, decrease of $L_\lambda^*(x)$ is

$$L_\lambda^*(x_0) - L_\lambda^*(x_k) \geq \frac{LS_k}{32k} - \frac{k\sigma^2}{2L} - \sigma\sqrt{kS_k} \geq \frac{LS_k}{100k}$$

$$\geq \frac{1}{100} L^{-\frac{1}{2}} \left(\frac{64\epsilon}{64c+1}\right)^{\frac{4+3\nu_f}{2+2\nu_f}} k H_\nu^{-\frac{1}{2+2\nu_f}}.$$

Sum above inequality over all epochs and denote the number of total iterates as $K$, we have

$$K \leq 100\Delta_\lambda L^{\frac{1}{2}} H_\nu^{\frac{1}{2+2\nu_f}} \left(\frac{64c+1}{64\epsilon}\right)^{\frac{4+3\nu_f}{2+2\nu_f}}. \tag{37}$$

As a result, we can denote the expression in the right side of equation 37 as $K_{\max}$. Substitute $H_\nu = \lambda^{\nu_f(1-\nu_g)}\mathcal{O}\left(\ell\kappa^{3+(1+\nu_g)\nu_f}\right)$ and $L = \mathcal{O}(\ell\kappa^3)$ for (37), we have

$$K \leq \mathcal{O}\left(\Delta_\lambda \lambda^{\frac{\nu_f(1-\nu_g)}{(2+2\nu_f)}} \ell^{\frac{2+\nu_f}{2+2\nu_f}} \kappa^{\frac{6+4\nu_f+\nu_f\nu_g}{(2+2\nu_f)}} \epsilon^{-\frac{4+3\nu_f}{2+2\nu_f}}\right). \tag{38}$$

We can also bound $S_k$ as:

$$S_k = S_k^{\frac{2+\nu_f}{2+2\nu_f}} S_k^{\frac{\nu_f}{2+2\nu_f}} \geq L^{-1} \left(\frac{64\epsilon}{64c+1}\right)^{\frac{2+\nu_f}{1+\nu_f}} k H_\nu^{-\frac{1}{1+\nu_f}}.$$

From Lemma 8, (34), (35), in this epoch, decrease of $L_\lambda^*(x)$ is

$$L_\lambda^*(x_0) - L_\lambda^*(x_k) \geq \frac{LS_k}{32k} - \frac{k\sigma^2}{2L} - \sigma\sqrt{kS_k}$$

$$\geq \frac{LS_k}{100k}$$

$$\geq \frac{1}{100} \left(\frac{64\epsilon}{64c+1}\right)^{\frac{2+\nu_f}{1+\nu_f}} H_\nu^{-\frac{1}{1+\nu_f}}. \tag{39}$$

Sum above inequalities over all epochs, we have

$$T \leq 100\Delta_\lambda \left(\frac{64c+1}{64\epsilon}\right)^{\frac{2+\nu_f}{1+\nu_f}} H_\nu^{\frac{1}{1+\nu_f}}. \tag{40}$$

Substitute $H_\nu = \lambda^{\nu_f(1-\nu_g)}\mathcal{O}\left(\ell\kappa^{3+(1+\nu_g)\nu_f}\right)$ and $L = \mathcal{O}(\ell\kappa^3)$ for equation 40, we have

$$T \leq \mathcal{O}\left(\Delta_\lambda \lambda^{\frac{\nu_f(1-\nu_g)}{(1+\nu_f)}} \ell^{\frac{1}{1+\nu_f}} \kappa^{\frac{3+(1+\nu_g)\nu_f}{(1+\nu_f)}} \epsilon^{-\frac{2+\nu_f}{1+\nu_f}}\right). \tag{41}$$

$\square$

### E.7 PROOF OF THEOREM 1

*Proof.* From Lemma 1, we have $\|\nabla L_\lambda^*(x) - \nabla\varphi(x)\| \leq \mathcal{O}(\ell\kappa^3)/\lambda$. From Lemma 1, we have $|L_\lambda^*(x) - \varphi(x)| \leq \mathcal{O}(\kappa^2)/\lambda$. Denote the number of total iterates as $K$, from Proposition 1, the following holds:

$$\|\nabla\varphi(\bar{w}_k)\| \leq \|\nabla L_\lambda^*(\bar{w}_k) - \nabla\varphi(\bar{w}_k)\| + \|\nabla L_\lambda^*(\bar{w}_k)\| \leq 2\epsilon.$$

Substitute equation 38 and equation 41 with $\lambda = \max(\mathcal{O}(\kappa), \mathcal{O}(\ell\kappa^3)/\epsilon, \mathcal{O}(\ell\kappa^2)/\Delta)$, the theorem is proved. $\square$

### E.8 PROOF OF THEOREM 2

**Lemma E.2.** *Consider the $t$-epoch generated by Algorithm 1 and ending at iteration $k$, we claim that for any $t$ and its corresponding $k$, we can find some constant $C$ to satisfy:*

$$\|\nabla L_\lambda \left(w_{t,k-1}\right)\|_2 \leq C.$$

*Proof.* For the $t$-epoch except the last epoch, $\bar{w}_{t,k}$ is not an $\epsilon$-first-order stationary point. Since $L_\lambda^*(x)$ has $L$-Lipschitz continuous gradient, we have

$$L_\lambda^* \left(x_{k+1}\right) \leq L_\lambda^* \left(w_k\right) + \left\langle \nabla L_\lambda^* \left(w_k\right), x_{k+1} - w_k \right\rangle + \frac{L}{2} \|x_{k+1} - w_k\|^2$$

$$\leq L_\lambda^* \left(w_k\right) - \frac{1}{L} \left\langle \nabla L_\lambda^* \left(w_k\right), \hat{\nabla} L_\lambda^* \left(w_k\right) \right\rangle + \frac{1}{2L} \left\|\hat{\nabla} L_\lambda^* \left(w_k\right)\right\|^2,$$

where we use $x_{k+1} = w_k - \frac{1}{L} \hat{\nabla} L_\lambda^* (w_k)$. We also have

$$L_\lambda^* \left(x_k\right) \geq L_\lambda^* \left(w_k\right) + \left\langle \nabla L_\lambda^* \left(w_k\right), x_k - w_k \right\rangle - \frac{L}{2} \|x_k - w_k\|^2.$$

Combining the above inequalities leads to

$$L_\lambda^* \left(x_{k+1}\right) - L_\lambda^* \left(x_k\right)$$

$$\leq - \left\langle \nabla L_\lambda^* \left(w_k\right), x_k - w_k \right\rangle + \frac{L}{2} \|x_k - w_k\|^2 - \frac{1}{L} \left\langle \nabla L_\lambda^* \left(w_k\right), \hat{\nabla} L_\lambda^* \left(w_k\right) \right\rangle + \frac{1}{2L} \left\|\hat{\nabla} L_\lambda^* \left(w_k\right)\right\|^2$$

$$= L \left\langle x_{k+1} - w_k, x_k - w_k \right\rangle + \left\langle \hat{\nabla} L_\lambda^* \left(w_k\right) - \nabla L_\lambda^* \left(w_k\right), x_k - w_k \right\rangle + \frac{L}{2} \|x_k - w_k\|^2$$

$$\quad - \frac{1}{L} \left\langle \nabla L_\lambda^* \left(w_k\right), \hat{\nabla} L_\lambda^* \left(w_k\right) \right\rangle + \frac{1}{2L} \left\|\hat{\nabla} L_\lambda^* \left(w_k\right)\right\|^2$$

$$= \frac{L}{2} \left( \|x_{k+1} - w_k\|^2 + \|x_k - w_k\|^2 - \|x_{k+1} - x_k\|^2 \right) + \left\langle \hat{\nabla} L_\lambda^* \left(w_k\right) - \nabla L_\lambda^* \left(w_k\right), x_k - w_k \right\rangle$$

$$\quad + \frac{L}{2} \|x_k - w_k\|^2 - \frac{1}{L} \left\langle \nabla L_\lambda^* \left(w_k\right), \hat{\nabla} L_\lambda^* \left(w_k\right) \right\rangle + \frac{1}{2L} \left\|\hat{\nabla} L_\lambda^* \left(w_k\right)\right\|^2$$

$$\leq L \|x_k - w_k\|^2 - \frac{L}{2} \|x_{k+1} - x_k\|^2 + \left\langle \hat{\nabla} L_\lambda^* \left(w_k\right) - \nabla L_\lambda^* \left(w_k\right), x_k - w_k \right\rangle + \frac{1}{L} \left\|\hat{\nabla} L_\lambda^* \left(w_k\right)\right\|^2$$

$$\quad - \frac{1}{L} \left\langle \hat{\nabla} L_\lambda^* \left(w_k\right), \nabla L_\lambda^* \left(w_k\right) \right\rangle$$

$$\overset{(a)}{\leq} L \|x_k - x_{k-1}\|^2 - \frac{L}{2} \|x_{k+1} - x_k\|^2 + \left\|\hat{\nabla} L_\lambda^* \left(w_k\right) - \nabla L_\lambda^* \left(w_k\right)\right\| \cdot \|x_k - x_{k-1}\|$$

$$\quad + \frac{1}{L} \left\|\hat{\nabla} L_\lambda^* \left(w_k\right)\right\|^2 - \frac{1}{L} \left\langle \nabla L_\lambda^* \left(w_k\right), \hat{\nabla} L_\lambda^* \left(w_k\right) \right\rangle$$

$$= L \|x_k - x_{k-1}\|^2 - \frac{L}{2} \|x_{k+1} - x_k\|^2 + \left\|\hat{\nabla} L_\lambda^* \left(w_k\right) - \nabla L_\lambda^* \left(w_k\right)\right\| \cdot \|x_k - x_{k-1}\|$$

$$\quad + \frac{1}{L} \left\|\hat{\nabla} L_\lambda^* \left(w_k\right)\right\|^2 - \frac{1}{2L} \left( \|\nabla L_\lambda^* \left(w_k\right)\|^2 + \left\|\hat{\nabla} L_\lambda^* \left(w_k\right)\right\|^2 - \left\|\nabla L_\lambda^* \left(w_k\right) - \hat{\nabla} L_\lambda^* \left(w_k\right)\right\|^2 \right)$$

$$\overset{(b)}{\leq} L \|x_k - x_{k-1}\|^2 - \frac{L}{2} \|x_{k+1} - x_k\|^2 + \left\|\hat{\nabla} L_\lambda^* \left(w_k\right) - \nabla L_\lambda^* \left(w_k\right)\right\| \cdot \|x_k - x_{k-1}\|$$

$$\quad - \frac{1}{4L} \|\nabla L_\lambda^* \left(w_k\right)\|^2 + \frac{3}{4L} \left\|\nabla L_\lambda^* \left(w_k\right) - \hat{\nabla} L_\lambda^* \left(w_k\right)\right\|^2$$

$$\overset{(c)}{\leq} L \|x_k - x_{k-1}\|^2 - \frac{L}{2} \|x_{k+1} - x_k\|^2 - \frac{1}{4L} \|\nabla L_\lambda^* \left(w_k\right)\|^2 + \sigma \|x_k - x_{k-1}\| + \frac{3}{4L} \sigma^2,$$

where we use $\|x_k - w_k\| = \theta_k \|x_k - x_{k-1}\| \leq \|x_k - x_{k-1}\|$ in $\overset{(a)}{\leq}$, the triangle inequality in $\overset{(b)}{\leq}$ and Lemma 4 in $\overset{(c)}{\leq}$.

Summing over the above inequality, and using $x_0 = x_{-1}$, we have

$$L_\lambda^* (x_k) - L_\lambda^* (x_0)$$

$$\leq \frac{L}{2} \sum_{i=0}^{k-2} \|x_{i+1} - x_i\|^2 - \frac{1}{4L} \sum_{i=0}^{k-1} \|\nabla L_\lambda^* (w_i)\|^2 + \sigma \sum_{i=0}^{k-1} \|x_i - x_{i-1}\| + \frac{3}{4L} \sigma^2 k$$

$$\overset{(d)}{\leq} \frac{L}{2} \sum_{i=0}^{k-2} \|x_{i+1} - x_i\|^2 - \frac{1}{4L} \sum_{i=0}^{k-1} \|\nabla L_\lambda^* (w_i)\|^2 + \sigma \sqrt{k-1} \sqrt{\sum_{i=0}^{k-2} \|x_{i+1} - x_i\|^2} + \frac{3}{4L} \sigma^2 k$$

$$\overset{(e)}{\leq} \frac{L}{2} S_{k-1} - \frac{1}{4L} \|\nabla L_\lambda^* (w_{k-1})\|^2 + \sigma \sqrt{k S_{k-1}} + \frac{3}{4L} \sigma^2 k$$

$$\overset{(f)}{\leq} \frac{L}{2} \left( \frac{L}{H_\nu} \right)^{\frac{2}{\nu_f}} - \frac{1}{4L} \|\nabla L_\lambda^* (w_{k-1})\|^2 + \sigma((L/H_\nu)^{\frac{1}{\nu_f}}) + \frac{3}{4L} \sigma^2 k$$

$$\overset{(g)}{\leq} \frac{L}{2} \left( \frac{L}{H_\nu} \right)^{\frac{2}{\nu_f}} - \frac{1}{4L} \|\nabla L_\lambda^* (w_{k-1})\|^2 + \sigma((L/H_\nu)^{\frac{1}{\nu_f}}) + \frac{3L S_k}{4 \times 64^2 k}, \tag{42}$$

where we use the Cauchy–Schwarz inequality in $\overset{(d)}{\leq}$, non-negativity of norm in $\overset{(e)}{\leq}$, the restart condition equation 5 in $\overset{(f)}{\leq}$ and equation 35 in $\overset{(g)}{\leq}$. For the last term in equation 42, we have

$$\frac{S_k}{k} \leq \frac{S_{k-1}}{k} + \frac{\|x_k - x_{k-1}\|^2}{k}$$

$$\overset{(a)}{\leq} \left( \frac{L}{H_\nu} \right)^{2/\nu_f} + \frac{\|x_k - x_{k-1}\|^2}{k}$$

$$\overset{(b)}{\leq} \left( \frac{L}{H_\nu} \right)^{2/\nu_f} + \frac{1}{k} \left\| w_{k-1} - x_{k-1} - \frac{1}{L} \hat{\nabla} L_\lambda^* (w_{k-1}) \right\|^2$$

$$\overset{(c)}{\leq} \left( \frac{L}{H_\nu} \right)^{2/\nu_f} + \frac{2}{k} \|w_{k-1} - x_{k-1}\|^2 + \frac{2}{k L^2} \left\| \hat{\nabla} L_\lambda^* (w_{k-1}) \right\|^2$$

$$\overset{(d)}{\leq} \left( \frac{L}{H_\nu} \right)^{2/\nu_f} + \frac{8}{k} \mathcal{D}^2 + \frac{4}{k L^2} \|\nabla L_\lambda^* (w_{k-1})\|^2 + \frac{4\sigma^2}{L^2},$$

where we use the restart condition equation 5 in $\overset{(a)}{\leq}$, $x_k = w_{k-1} - \frac{1}{L} \hat{\nabla} L_\lambda^* (w_{k-1})$ in $\overset{(b)}{\leq}$, Lemma 3 in $\overset{(c)}{\leq}$ and Lemma 4 in $\overset{(d)}{\leq}$. Combined with equation 42, we obtain

$$L_\lambda^* (x_k) - L_\lambda^* (x_0)$$

$$\leq \left( \frac{1}{2} + \frac{3}{4 \times 64^2} \right) L \left( \frac{L}{H_\nu} \right)^{\frac{2}{\nu_f}} + \frac{3L}{4 \times 64^2} \left( \frac{8}{k} \mathcal{D}^2 + \frac{4\sigma^2}{L^2} \right) \tag{43}$$

$$- \left( \frac{1}{4L} - \frac{3}{64^2 L} \right) \|\nabla L_\lambda^* (w_{k-1})\|^2 + \sigma((L/H_\nu)^{\frac{1}{\nu_f}})$$

We claim that for any $t$-th epoch ending at iteration $k$, we can find some constant $C$ to satisfy:

$$\|\nabla L_\lambda (w_{t,k-1})\|_2 \leq C.$$

Otherwise, equation 43 shows that $L_\lambda^* (w_{t,k})$ can go to $-\infty$, which contradicts to $\min_{x \in \mathbf{R}^{d_x}} \varphi(x) > -\infty$ in Assumption 1 and $|L_\lambda^*(x) - \varphi(x)| \leq \mathcal{O}(\ell \kappa^2 / \lambda)$ in Lemma 1. $\qquad \square$

With the help of Lemma E.2, we provide the proof of Theorem 2.

*Proof.* We firstly show the boundedness of $\|y^*(w_{t,0})\|$. Suppose that the $t$-epoch ends at iteration $k$, we have

$$
\begin{aligned}
&\|y^*(w_{t+1,0}) - y^*(w_{0,0})\| \\
\leq& \|y^*(x_{t,k}) - y^*(w_{t,k-1})\| + \|y^*(w_{t,k-1}) - y^*(w_{t,0})\| + \|y^*(w_{t,0}) - y^*(w_{0,0})\| \\
\leq& \frac{L_g}{\mu}\|x_{t,k} - w_{t,k-1}\| + \frac{L_g}{\mu}\|w_{t,k-1} - w_{t,0}\| + \|y^*(w_{t,0}) - y^*(w_{0,0})\| \\
\leq& \frac{L_g}{\mu}(\frac{C+\sigma}{L} + \mathcal{D}) + \|y^*(w_{t,0}) - y^*(w_{0,0})\|.
\end{aligned}
$$

The first inequality holds due to triangular inequality, the second inequality holds due to $y^*(x)$ is $L_g/\mu$-Lipschitz continuous and the last inequality holds due to Lemma 4 and Lemma E.2. Then we have

$$
\begin{aligned}
\|y^*(w_{t,0})\| \leq& \|y^*(w_{t,0}) - y^*(w_{0,0})\| + \|y^*(w_{0,0})\| \\
\leq& \|y^*(w_{0,0})\| + \frac{L_g}{\mu}(\frac{C+\sigma}{L} + \mathcal{D})t \\
\leq& \|y^*(w_{0,0})\| + \frac{L_g}{\mu}(\frac{C+\sigma}{L} + \mathcal{D})T,
\end{aligned}
$$

where $T$ is the total number of epochs. We can set $\{T_{t,i}, T'_{t,i}\}$ as follows: let

$$
T_{t,i} = \left\lceil 2\sqrt{\frac{L_g}{\mu}} \log \sqrt{1 + \frac{L_g}{\mu}\left(1 + 2\lambda\frac{L_g^2}{\sigma\mu}(\frac{C+\sigma}{L} + 5\mathcal{D})\right)} \right\rceil, \tag{44}
$$

$$
T'_{t,i} = \left\lceil 2\sqrt{\frac{4L_g}{\mu}} \log \sqrt{1 + \frac{4L_g}{\mu}\left(1 + 16\lambda\frac{L_g^2}{\sigma\mu}(\frac{C+\sigma}{L} + 5\mathcal{D})\right)} \right\rceil \tag{45}
$$

for $i \geq 1$, and

$$
T_{t,i} = \left\lceil 2\sqrt{\frac{L_g}{\mu}} \log \sqrt{1 + \frac{L_g}{\mu}\left(\|y^*(w_{0,0})\| + \frac{L_g}{\mu}(\frac{C+\sigma}{L} + \mathcal{D})T\right)\frac{2\lambda L_g}{\sigma}} \right\rceil, \tag{46}
$$

$$
T'_{t,i} = \left\lceil 2\sqrt{\frac{4L_g}{\mu}} \log \sqrt{1 + \frac{4L_g}{\mu}\left(\|y^*(w_{0,0})\| + \frac{4L_g}{\mu}(\frac{C+\sigma}{L} + \mathcal{D})T\right)\frac{4\lambda L_g}{\sigma}} \right\rceil \tag{47}
$$

for $i = 0$, where $T$ is the total number of epochs. From Theorem 1, we know that

$$
T \leq \mathcal{O}(\Delta\ell^{\frac{1+\nu_f-\nu_f\nu_g}{1+\nu_f}} \kappa^{\frac{3+4\nu_f-2\nu_f\nu_g}{1+\nu_f}} \epsilon^{-\frac{2+2\nu_f-\nu_f\nu_g}{1+\nu_f}}).
$$

Then we prove equation 8 holds for $z_{t,i}$ by induction. For $i = 0$, by the definition of $T_{t,0}$ in equation 46, we have

$$
\begin{aligned}
\|z_{t,0} - y^*(w_{t,0})\| \leq& \sqrt{1 + \frac{L_g}{\mu}}(1 - \sqrt{\frac{\mu}{L_g}})^{T_{t,0}/2}\|z_{t,-1} - y^*(w_{t,0})\| \\
\leq& \sqrt{1 + \frac{L_g}{\mu}}(1 - \sqrt{\frac{\mu}{L_g}})^{T_{t,0}/2}\|y^*(w_{t,0})\| \\
\leq& \frac{\sigma}{2\lambda L_g}.
\end{aligned}
$$

From Lemma D.1, if $i \geq 1$, we have

$$\|z_{t,i} - y^*(w_{t,i})\| \leq \sqrt{1 + \frac{L_g}{\mu}}(1 - \sqrt{\frac{\mu}{L_g}})^{T_{t,i}/2}\|z_{t,i-1} - y^*(w_{t,i})\|$$

$$\overset{(a)}{\leq} \sqrt{1 + \frac{L_g}{\mu}}(1 - \sqrt{\frac{\mu}{L_g}})^{T_{t,i}/2}\left(\|y^*(w_{t,i}) - y^*(w_{t,i-1})\| + \|z_{t,i-1} - y^*(w_{t,i-1})\|\right)$$

$$\overset{(b)}{\leq} \sqrt{1 + \frac{L_g}{\mu}}(1 - \sqrt{\frac{\mu}{L_g}})^{T_{t,i}/2}\left(\frac{L_g}{\mu}\|w_{t,i} - w_{t,i-1}\| + \frac{\sigma}{2\lambda L_g}\right)$$

$$\overset{(c)}{\leq} \sqrt{1 + \frac{L_g}{\mu}}(1 - \sqrt{\frac{\mu}{L_g}})^{T_{t,i}/2}\left(\frac{2L_g}{\mu}\|x_{t,i} - x_{t,i-1}\| + \frac{L_g}{\mu}\|x_{t,i-1} - x_{t,i-2}\| + \frac{\sigma}{2\lambda L_g}\right)$$

$$\overset{(d)}{\leq} \sqrt{1 + \frac{L_g}{\mu}}(1 - \sqrt{\frac{\mu}{L_g}})^{T_{t,i}/2}\left(\frac{L_g}{\mu}(\frac{C + \sigma}{L} + 5\mathcal{D}) + \frac{\sigma}{2\lambda L_g}\right)$$

$$\overset{(e)}{\leq} \frac{\sigma}{2\lambda L_g},$$

where the inequality $\overset{(a)}{\leq}$ follows from the triangle inequality, $\overset{(b)}{\leq}$ uses the inductive hypothesis and the fact that $y^*(x)$ is $L_g/\mu$-Lipschitz continuous, $\overset{(c)}{\leq}$ holds by the definition $w_{t,i} = x_{t,i} + \theta_i(x_{t,i} - x_{t,i-1})$, $\overset{(d)}{\leq}$ applies Lemma 3 and Lemma E.2, and $\overset{(e)}{\leq}$ follows from equation 44. Therefore, by mathematical induction, we conclude that equation 8 holds for all $z_{t,i}$ with $\{T_{t,i}\}$ defined in equation 44, equation 46. Similarly, we can prove that equation 8 holds for $y_{t,i}$ with $T'_{t,i}$ defined in equation 45, equation 47. So all $y_{t,i}$ and $z_{t,i}$ satisfy Condition 1. The total first-order oracle complexity is $\sum_{t,i} T_{t,i}$, i.e.,

$$\tilde{\mathcal{O}}\left(\Delta\ell^{\frac{2+2\nu_f - \nu_f\nu_g}{2+2\nu_f}}\kappa^{\frac{7+8\nu_f - 2\nu_f\nu_g}{2+2\nu_f}}\epsilon^{-\frac{4+4\nu_f - \nu_f\nu_g}{2+2\nu_f}}\right).$$

When $\nu_f = \nu_g = 1$, the first-order oracle complexity is $\tilde{\mathcal{O}}\left(\Delta\ell^{3/4}\kappa^{13/4}\epsilon^{-7/4}\right)$.

$\square$