# OpenReview forum: "Accelerating first-order methods for nonconvex-strongly-convex bilevel optimization under general smoothness"
_ICLR.cc/2026/Conference — ICLR 2026 Conference Withdrawn Submission_

### Official Review · Reviewer_8Fo8 · 2025-11-01

**Soundness:** 2
**Presentation:** 2
**Contribution:** 2
**Rating:** 2
**Confidence:** 4

**Summary:**

The paper proposes RAGD-GS, a restarted accelerated first-order framework for nonconvex–strongly-convex bilevel optimization that works through a penalized surrogate $L_{\lambda}^{\*}(x)$ and an inexact hyper-gradient estimator computed by two inner AGD solvers (for $y^\*(x)$ and $y_{\lambda}^{*}(x)$). It claims an accelerated complexity of $\tilde{O}(\epsilon^{-7/4})$ in the Lipschitz case while extending to Hölder-continuous higher-order smoothness, and reports small-scale experiments on data hyper-cleaning and hyperparameter optimization.

**Strengths:**

- The RAGD outer loop plus two inner AGDs is easy to implement, and the estimator of $\nabla L_{\lambda}^{*}$ is explicitly stated and purely first-order.

- Restart condition, potential-function descent, and complexity theorems are self-contained and align with recent nonconvex AGD analyses.

- Experiments compare against AID/ITD/BA/RAHGD/F2BA/AccF2BA and support the theoretical convergence guarantees.

**Weaknesses:**

- The paper claims to improve the first-order oracle complexity to $\tilde{O}(\ell^{3/4}\kappa^{13/4}\epsilon^{-7/4})$ without requiring second-order oracles. However, this result has already been achieved by the AccF$\^2$BA method in prior work [1] using first-order oracles only.

- While this work considers the general Hölder continuity setting, it does not empirically explore or validate the Hölder case in Section 5.

- Condition 1 and Theorem 2 require inner solves tight enough that costs are hidden in $\tilde{O}$, but the experiments do not report the actual inner iteration counts that satisfy (8).



[1] Chen, L., Ma, Y., & Zhang, J. (2025). Near-optimal nonconvex-strongly-convex bilevel optimization with fully first-order oracles. Journal of Machine Learning Research, 26(109), 1-56.

**Questions:**

- How do you estimate $H_{\nu}$ and $D$ to run the restart test in equation (5) without oracle knowledge in experiments?

- Regarding the Hölder regime, could you include synthetic experiments where $\nu_f, \nu_g < 1$ (controlled via constructed objectives) to empirically validate the exponents predicted in Theorem 1?

---

### Official Review · Reviewer_UCqE · 2025-11-01

**Soundness:** 2
**Presentation:** 3
**Contribution:** 2
**Rating:** 4
**Confidence:** 3

**Summary:**

This paper proposes an "accelerated first-order" framework for nonconvex–strongly-convex bilevel optimization under generalized Holder continuity assumptions on higher-order derivatives. The method combines the penalty-based bilevel method with a restarted accelerated gradient descent (RAGD-GS). The proposed algorithm achieves an $\tilde{O}(\ell^{3/4}\kappa^{13/4}\epsilon^{-7/4})$ first-order oracle complexity, matching the state-of-the-art accelerated results.

**Strengths:**

- The paper provides a technically correct extension of prior “fully first-order” acceleration frameworks to a slightly more general smoothness class (Hölder instead of Lipschitz).
- The convergence rate can cover match the state-of-the-art accelerated results under the standard Lipschitz continuity setting.

**Weaknesses:**

- This paper lacks technique novelty for bilevel optimization problem, and the entire framework is a combination of existing papers:
    - The penalty-surrogate method and its gradient construction are directly inherited from [F$^2$SA](https://arxiv.org/pdf/2301.10945) and [F$^2$BA](https://www.jmlr.org/papers/volume26/23-1104/23-1104.pdf).
    - The restart mechanism is essentially a parameterized variant of the [RAHGD](https://arxiv.org/pdf/2307.00126) restart rule, rewritten to include a Hölder-dependent constant instead of a fixed Lipschitz threshold.
    - The complexity result reproduces $\tilde{O}(\epsilon^{-7/4})$ rate already established by F$^2$BA/AccF$^2$BA.

- The new assumption "Holder continuous in higher-order derivatives" is not a relaxation for the prior assumption in F$^2$SA. In my mind, the new assumptions are different, not weaker. They replace the standard third-order derivative Lipschitz conditions with Holder-continuous condition but still require bounded constants that are functionally equivalent for practical analysis. The proof is almost the same when handling the higher-oder smoothness. They don't provide theoretical or empirical evidence that the Holder setting broadens the applicability to any class of bilevel problems previously excluded.

- The experimental section provides no empirical justification for the new assumptions. The proposed algorithm achieves nearly identical performance to AccF$^2$BA in Figure 3.

- There is no stochastic analysis. In real bilevel problems—hyperparameter tuning, meta-learning, or data hypercleaning, gradients are noisy or sampled.

Above all, the paper does not provide substantive theoretical or empirical advancement over existing first-order bilevel acceleration methods.

**Questions:**

Please check Weaknesses

---

### Official Review · Reviewer_U2GS · 2025-11-01

**Soundness:** 2
**Presentation:** 2
**Contribution:** 2
**Rating:** 2
**Confidence:** 4

**Summary:**

This paper proposes a restarted accelerated gradient method for nonconvex-strongly-convex bilevel optimization problems under generalized Hölder continuity assumptions. The method builds upon a penalty-based reformulation to create a single-loop algorithm that uses only first-order oracles. The authors provide a theoretical analysis showing an improved oracle complexity of $\tilde{\mathcal{O}}(\epsilon^{-7/4})$ under Lipschitz continuity, matching concurrent work, and more general rates under Hölder continuity.

**Strengths:**

1. The paper extends the analysis of accelerated bilevel methods from the standard Lipschitz continuity setting to the more general Hölder continuity framework.

2. The proposed RAGD-GS is a fully first-order method, avoiding the need for computationally expensive Hessian-vector or Jacobian-vector product oracles.

**Weaknesses:**

1. The primary claimed contribution of the paper is weakening the smoothness requirements for bilevel optimization from Lipschitz continuity of second- and third-order derivatives to Hölder continuity. However:

There already exist bilevel optimization methods that do not assume Lipschitz continuity of
$\nabla^2 f$ or $\nabla^3 g$, such as [14] and the following relevant reference omitted by the authors:
Chen, Xuxing, Tesi Xiao, and Krishnakumar Balasubramanian. "Optimal algorithms for stochastic bilevel optimization under relaxed smoothness conditions." JMLR 25.151 (2024): 1–51.

The paper does not sufficiently motivate why Hölder continuity is important in machine learning contexts. Are there common, practical ML applications where Hölder smoothness holds but Lipschitz smoothness does not?

There is no empirical evaluation demonstrating value in the Hölder setting. As a result, the contribution risks appearing as a purely theoretical generalization without practical relevance.


2. The $\tilde{O}(\epsilon^{-7/4})$ rate (Theorem 2, for $\nu_f=\nu_g=1$) matches the rate of AccF2BA [13] exactly, as shown in Table 1. The authors acknowledge this, stating their result is "consistent with the concurrent findings of [13]". This means the paper essentially proposes a different algorithm (using a different restart scheme) to achieve the same result that is already present in the literature. The contribution is therefore methodological (a different path) rather than a new state-of-the-art complexity, which significantly lessens its impact.


3. The algorithmic components (penalty formulation, AGD subroutines, restart condition) are not novel. The primary novelty lies in the theoretical analysis under a different set of assumptions. The restart condition, while different from prior work, is not motivated by a clear, fundamental insight that leads to a qualitative improvement.


4. The experiments in Figures 1, 2, and 3 are not compelling. They show that the proposed RAGD-GS performs very similarly to AccF2BA [13] and RAHGD [15]. This is not surprising, as it shares the same theoretical rate as AccF2BA. The experiments merely confirm that the algorithm works as expected, but they do not demonstrate any practical advantage over the existing state-of-the-art. Given that the main result is not novel, stronger empirical performance (e.g., significantly better constants, more stability) would be needed to justify the contribution, but this is not shown.

**Questions:**

1. Can you provide a clear example of a practical bilevel optimization problem in machine learning that fails the Lipschitz smoothness assumption but satisfies the Hölder one, where your generalized theory would be necessary?

2. Since your complexity under Lipschitz assumptions is identical to [13], what is the practical, empirical, or theoretical advantage of your proposed restart scheme (Eq. 5) over the one used in AccF2BA [13]? Your experiments do not seem to show a clear difference.

---

### Official Review · Reviewer_KzPz · 2025-11-03

**Soundness:** 3
**Presentation:** 3
**Contribution:** 2
**Rating:** 2
**Confidence:** 4

**Summary:**

The paper studies nonconvex–strongly-convex bilevel optimization problems and proposes an accelerated first-order framework under general (Hölder-continuous) smoothness assumptions. The authors introduce a penalty reformulation that removes the need for exact implicit differentiation and combine it with a restarted accelerated gradient descent scheme (RAGD-GS). The analysis establishes an improved first-order oracle complexity of $\tilde O(\varepsilon^{-7/4})$ in the Lipschitz-smooth case, extending acceleration theory to bilevel settings. Empirical validation on the MNIST hypercleaning task demonstrates faster convergence than existing bilevel algorithms.

**Strengths:**

The theoretical development is rigorous and technically strong. The paper unifies several existing bilevel frameworks under a single smoothness-generalized setting and achieves provable acceleration. The presentation of the penalized objective and the restart mechanism is clear and mathematically elegant. The convergence proofs are complete and well organized.

**Weaknesses:**

The analysis assumes deterministic nonconvex–strongly-convex regime and exact gradients in both levels.

The paper does not extend to stochastic or weakly convex inner problems, which limits its relevance for large-scale or deep learning applications.

The complexity bound hides heavy dependence on the condition number $\kappa$ and smoothness constants, which may offset the practical benefit of the improved $\varepsilon$-rate.

Empirically, only the data-hypercleaning benchmark on MNIST is presented, so the experimental evidence remains narrow.

**Questions:**

1. The analysis assumes strong convexity of the lower-level objective $g(x, y)$. Could the authors clarify whether the acceleration guarantees extend to weakly convex or nonconvex inner problems?
2. The penalty parameter $\lambda$ plays a central role in the convergence proof. How should it be selected in practice, and is there any adaptive rule ensuring stability without prior knowledge of problem constants?
3. The improved complexity $\tilde{O}(\varepsilon^{-7/4})$ depends on Hölder smoothness. Could the authors specify the exact constants hidden in the $\tilde{O}$ notation and discuss their dependence on the condition number $\kappa$?
4. The restart condition for the RAGD-GS algorithm is theoretically motivated. How sensitive is the method to the choice of restart interval or stopping criterion in empirical implementations?
5. The analysis is deterministic. Can the authors comment on whether similar acceleration rates could be achieved under stochastic gradients or sampling noise?
6. How does the proposed penalty formulation compare theoretically to implicit differentiation or Moreau-envelope-based bilevel methods in terms of approximation bias?
7. Are there lower bounds suggesting that $\tilde{O}(\varepsilon^{-7/4})$ is optimal for nonconvex–strongly-convex bilevel problems under the assumed smoothness conditions?
8. The empirical evaluation is limited to the MNIST hypercleaning task. Could the authors provide evidence that the acceleration behavior persists in higher-dimensional or large-scale settings?

---

### Note · Authors · 2025-11-18

I have read and agree with the venue's withdrawal policy on behalf of myself and my co-authors.